# A pharmacophore-guided deep learning approach for bioactive molecular generation

Huimin Zhu[1,5], Renyi Zhou[1,5], Dongsheng Cao[2], Jing Tang [ID][3,4] & Min Li [ID][1] ✉

The rational design of novel molecules with the desired bioactivity is a critical but challenging task in drug discovery, especially when treating a novel target family or understudied targets. We propose a Pharmacophore-Guided deep learning approach for bioactive Molecule Generation (PGMG). Through the guidance of pharmacophore, PGMG provides a flexible strategy for generating bioactive molecules. PGMG uses a graph neural network to encode spatially distributed chemical features and a transformer decoder to generate molecules. A latent variable is introduced to solve the many-to-many mapping between pharmacophores and molecules to improve the diversity of the generated molecules. Compared to existing methods, PGMG generates molecules with strong docking affinities and high scores of validity, uniqueness, and novelty. In the case studies, we use PGMG in a ligand-based and structure-based drug de novo design. Overall, the flexibility and effectiveness make PGMG a useful tool to accelerate the drug discovery process.

The acquisition of biologically active compounds is a vital but challenging step in drug discovery. It has been estimated that the drug-like chemical space is as large as $10^{60}$ for molecules obeying Lipinski's 'rule of five'[1,2], a set of criteria to evaluate a compound's potential to be an orally active drug based on its molecular properties. Therefore, it is extremely difficult to search for desired molecules in such a huge space. Traditionally, hit compounds that exhibit initial activity on a specific target could be obtained from natural products designed by medicinal chemists or acquired by high-throughput screening[3]. These methods consume a lot of human and financial resources, making the acquisition of hit compounds inefficient and costly. Recently, deep generative models have been proposed for the rational design of novel molecules with the desired properties, providing a new perspective for this task. Among the popular architectures and models for generating molecules from deep neural networks, the variational autoencoders[4,5], reinforcement learning[6,7], generative adversarial networks[8–10] and autoregressive models[11,12] have been successful in designing the desired molecules at a specified precondition. Many methods aim at generating molecules with given physicochemical properties, such as the Wildman–Crippen partition coefficient (LogP), synthetic

accessibility (SA), molecular weight (MW) and quantitative estimate of drug likeness (QED). However, a more practical and challenging objective is to design molecules that satisfy those properties involving biological experiments or extensive calculations to approximate, such as bioactivity of the molecules for a specific target[13]. To generate bioactive molecules, the existing models require a large dataset of known active molecules to fine-tune. However, this dataset may not be available. The paucity of activity data is one of the main obstacles in applying deep learning-based methods in drug design, especially for a novel target family. The choice of drug design strategy also depends on what information can be used, for example, the receptor structure or some known active ligands, hence narrowing down the application of many deep learning methods.

To overcome the problems of data scarcity, methods that combine prior biochemical knowledge into molecule generation models have been proposed. For example, conditioned generative adversarial network is used to design active-like molecules for inducing the desired gene expression signatures[14]. The Seq2Seq[15] method exploits a pretrained biochemical language model with two-stage fine-tuning to generate active-like molecules using the target protein sequence as the

[1]School of Computer Science and Engineering, Central South University, Changsha 410083, China. [2]Xiangya School of Pharmaceutical Sciences, Central South University, Changsha 410008, China. [3]Research Program in Systems Oncology, Faculty of Medicine, University of Helsinki, Helsinki 00290, Finland. [4]Department of Biochemistry and Developmental Biology, Faculty of Medicine, University of Helsinki, Helsinki 00290, Finland. [5]These authors contributed equally: Huimin Zhu, Renyi Zhou. ✉e-mail: limin@mail.csu.edu.cn

input. However, the structure–activity relationship for the molecules generated by such methods is less interpretable. DeLinker[16] and SyntaLinker[17] retain active fragments while updating linkers to generate active molecules. DEVELOP[18] combines DeLinker with chemical features as the constraints to improve the quality of the generated molecules. Fragment-based approaches require explicit knowledge of the active fragments, which may lead to a restricted chemical space for the model to explore. DeepLigBuilder[19], Pocket2Mol[20] and RELARION[21] generate molecules that are based on the binding sites between molecules and proteins in 3D Euclidean space. However, these methods are limited when the binding site or target structure is unknown. There are other methods that use chemical features in molecule generation, such as Reduced Graph[22], which simplifies a SMILES to an acyclic graph of functional group as its input. A shape-based method proposed by ref. 23 can generate molecules from a 3D representation of a seed ligand.

In the present study, we propose Pharmacophore-Guided deep learning approach for bioactive Molecule Generation (PGMG), a pharmacophore-guided molecule generation approach based on deep learning. PGMG uses pharmacophore hypotheses as a bridge to connect different types of activity data. Here, a pharmacophore is a set of spatially distributed chemical features necessary for a drug to bind to a target. Pharmacophore hypotheses can be constructed by superimposing a few active compounds[24] or can be inferred from the structure of a given target[25]. Pharmacophore-based drug design has many successful applications[26,27], but its potential in deep generative models has not been fully exploited. The aforementioned REALTION incorporates pharmacophore information but only as an auxiliary tool to assist in the generation based on complexes and active molecules. In PGMG, we provide a different approach that enables flexible generation without further fine-tuning in different drug design scenarios, especially for newly discovered targets where there is insufficient activity data. We use a complete graph to fully represent a pharmacophore, with each node corresponding to a pharmacophore feature, such that the spatial information can be encoded as the distance between each node pair. Using the graph as the sole input, PGMG can generate molecules that match the corresponding pharmacophore. This gives PGMG the capability to utilise different types of activity data in a uniform representation and biologically meaningful way to control the process of molecule design. Furthermore, since pharmacophores and molecules have a many-to-many relationship, PGMG introduces latent variables to model such a relationship to boost the variety of generated molecules. In addition, a transformer structure is employed as the backbone to learn the implicit rules of SMILES strings to map between latent variables and molecules. We comprehensively evaluate PGMG performance in molecule generation with goal-directed and drug-like metrics. The results show that PGMG can generate molecules satisfying the given pharmacophore hypotheses and pharmacokinetic requirements while maintaining a high level of validity, uniqueness, and novelty. The case studies further demonstrate that PGMG provides an effective strategy for ligand-based and structure-based drug de novo designs.

## Results

### Overview of PGMG

PGMG is a pharmacophore-guided molecular generation approach based on deep learning. The overall architecture of PGMG is illustrated in Fig. 1. The goal of PGMG is to generate molecules matching a given pharmacophore. Here, we introduce a set of latent variables $z$ to deal with the many-to-many mapping between pharmacophores and molecules. Namely, a molecule $x$ can be represented as a unique combination of two complementary encodings, including $c$, which represents the given pharmacophore, and $z$, which corresponds to how chemical groups are placed within the molecule. The latent

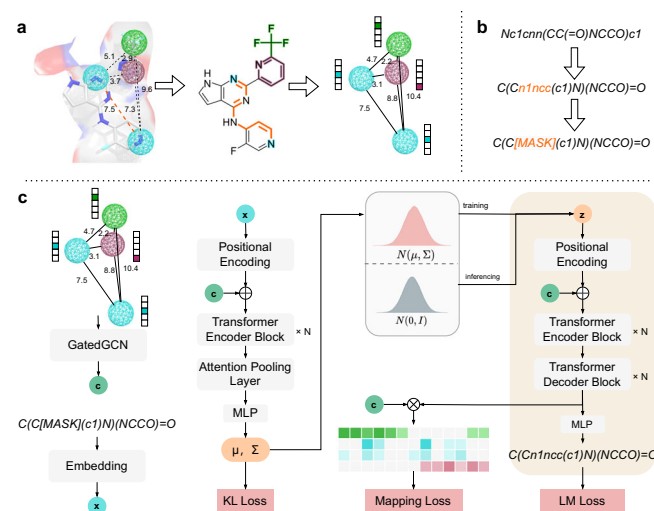

**Fig. 1 | The overall architecture of PGMG. a** Construction of the pharmacophore networks. We use the shortest paths on the molecular graph to determine the distances between two pharmacophore features, based on which a fully connected graph was constructed to represent a pharmacophore hypothesis. Different colours represent different types of pharmacophore features. **b** Preprocessing of SMILES. We randomise a given canonical SMILES and corrupt it using the infilling scheme. **c** Pipelines for model training and inference. $c$ represents the embedding vector sequences for the given pharmacophore hypothesis; $x$ represents the embedding sequence of the input SMILES; and $z$ represents the latent variables for a molecule. During inference, $z$ is drawn from a predefined normal distribution $N(0,I)$ while during training, it is sampled from a learned distribution $N(\mu,\Sigma)$. The transformer encoder and decoder blocks are stacked with N layers. $\oplus$ denotes the concatenation of two vectors and $\otimes$ matrix multiplication. The overlap between the training and inferencing processes is highlighted in the right panel. GatedGCN stands for Gated Graph Convolutional Network, and MLP stands for Multi-Layer Perceptron.

variable $z$ grants PGMG the ability to model multiple modes in the conditional distribution:

$$P(x|c) = \int_{z \sim P(z|c)} P(x|c,z)P(z|c)dz \qquad (1)$$

We train two neural networks, an encoder network $P_\phi(z|c,x)$ to approximate $P(z|c)$ indirectly and a decoder network $P_\theta(x|c,z)$ to approximate $P(x|c,z)$. We embed molecules in the SMILES format into dense feature vectors and use Gated GCN[28] to embed pharmacophore hypotheses. The transformer structure proposed by ref. 29 is used as the backbone of our model to learn the mapping between the pharmacophore and molecular structures.

A PGMG training sample can be constructed using the SMILES representation of a molecule. First, the chemical features of a molecule are identified using RDKit[30], some of which are randomly selected to build a pharmacophore network $G_p$. As shown in Fig. 1a, we use the shortest-path distances on the molecular graph to replace the Euclidean distances between two pharmacophore features in a pharmacophore hypothesis. The description of pharmacophore types can be found in Supplementary Table 1, and the mapping rule between the shortest-paths and Euclidean distances is available in Supplementary Table 2. The analysis of the correlation and differences between the shortest-path distance and Euclidean distance can be found in the Supplementary Information (Supplementary Figs. 1–4). Next, a molecule is represented as a randomised SMILES string and then segmented into a token sequence $s$. We then corrupt $s$ to obtain the encoder input $s'$ by using the infilling scheme[31] and obtaining a training sample $(G_p, s, s')$. Since we avoid the use of target-specific activity data in the

training stage, PGMG bypasses the problem of data scarcity on active molecules.

When using the trained model to generate molecules, a pharmacophore hypothesis is required. The generation process is as follows: given a pharmacophore hypothesis $c$, a set of latent variables $z$ is sampled from the prior distribution $p(z|c)$, which, in our case, is the standard Gaussian distribution $N(0,I)$, and the molecules are then generated from the conditional distribution $p(x|z,c)$. There are multiple ways to construct a pharmacophore using various types of active data.

We demonstrate the use of both ligand- and structure-based pharmacophores to generate active molecules for de novo drug design.

### Performance of PGMG on the unconditional molecule generation task

We evaluate our model's performance on the unconditional molecule generation task with other SMILES-based methods, including VAE[4], ORGAN[9], SMILES LSTM[32] and Syntalinker[17]. We have trained these models on the ChEMBL dataset[33] based on the train-test split used in the GuacaMol benchmark[34]. Since PGMG is a conditional model, we have approximated the unconditional distribution by generating molecules based on randomly sampled pharmacophore features. The molecule generation performance is evaluated by four metrics: validity, novelty, uniqueness and ratio of the available molecules (see the methods section for the definition of the metrics).

As shown in Table 1, PGMG performs the best in novelty and with the ratio of available molecules, while achieving a comparable level of validity and uniqueness as the other top models such as Syntalinker and SMILES LSTM. We consider the ratio of available molecules as a primary metric because it assesses the performance of the model in

generating novel molecules. Notably, PGMG has been found to improve the ratio of available molecules by 6.3%.

To test whether PGMG catches the distribution of the training datasets, we have further examined the physicochemical properties of the generated molecules. As shown in Fig. 2, physicochemical properties such as the MW, LogP, QED, and topological polar surface area (TPSA) share similar distributions between the generated and training molecules. This demonstrates that PGMG well captures the distribution of the molecules in the training dataset.

### PGMG can generate bioactive molecules satisfying the given pharmacophores

We have evaluated the extent to which the generated molecules fit the given pharmacophore hypotheses. Furthermore, we have predicted binding affinities between protein receptors and molecules that are generated using PGMG through the molecular docking tool AutoDock Vina[35].

We have used a match score to estimate the matching degree between each molecule-pharmacophore pair. The definition of the match score and an algorithm for calculating the match score can be seen in the Supplementary Information. To make the calculation process understandable, we give some examples about the calculation of the match score (Supplementary Fig. 5). We extract a random pharmacophore hypothesis from each molecule in the test dataset. About 236,000 molecules in total are generated from these random pharmacophore hypotheses. For comparison, we have also calculated the match score between 236,000 random molecules from the ChEMBL dataset[33] and the selected pharmacophores.

As shown in Fig. 3, most of the generated molecules (83.6%) have matching scores greater than 0.8, of which 78.6% have a matching score of 1.0. In contrast, the matching degrees for the random molecules are centred at 0.466, with only 4.91% having a matching score of 1.0. This result demonstrates PGMG's ability to generate molecules satisfying the given pharmacophore hypotheses.

To further examine the binding activity of molecules generated by PGMG through the guidance of pharmacophores, we obtain pharmacophore hypotheses with known target structures from the literature. For each pharmacophore hypothesis, 10,000 molecules are generated by PGMG, for which the docking scores are calculated by AutoDock Vina[35]. In Fig. 4a, we show the docking score distributions of the top 1000 molecules generated by PGMG and top 1000 molecules with known bioactivity for the 15 targets from the ChEMBL database. We have found that the molecules generated by PGMG obtain a comparable docking score with active molecules, suggesting that these molecules can bind to the 3D structure of targets (Supplementary Fig. 6 and Supplementary Table 4). We also conducted a

### Table 1 | Performance of PGMG and other SMILES-based models

| Methods | Validity↑ | Uniqueness↑ | Novelty↑ | Ratio of Available Molecules↑ |
|---|---|---|---|---|
| ORGAN[9] | 0.379 | 0.841 | 0.687 | 0.219 |
| VAE[4] | 0.870 | 0.999 | 0.974 | 0.847 |
| SMILES LSTM[32] | 0.959 | **1.000** | 0.912 | 0.875 |
| Syntalinker[17] | **1.000** | 0.880 | 0.903 | 0.795 |
| PGMG | 0.982 | 0.979 | **0.976** | **0.938** |

An upward arrow next to each metric indicates that higher values represent better performance. Best performance among all methods for each metric is shown in bold.

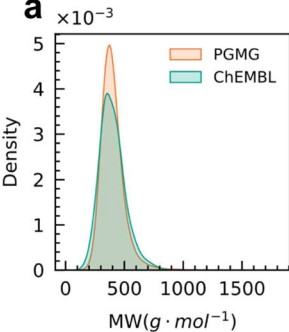 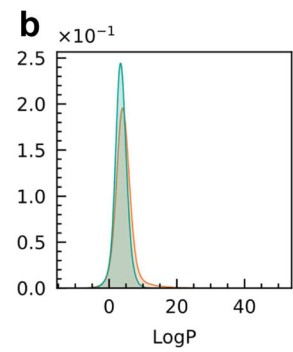 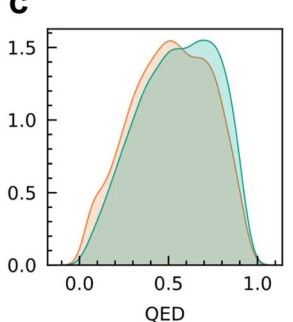 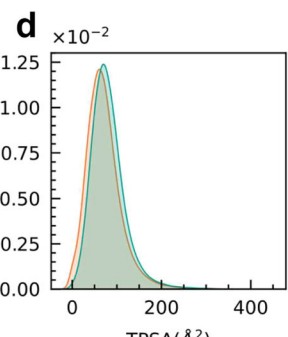

**Fig. 2 | Distribution of the physicochemical properties for the ChEMBL training set and molecules generated by PGMG. a** Molecule weight (MW); (**b**) the Wildman−Crippen partition coefficient (LogP); (**c**) quantitative estimate of drug-likeness (QED); (**d**) topological polar surface area (TPSA). The PGMG generated molecules include a total of 100,000 molecules from random pharmacophore hypotheses and the ChEMBL molecules comprise 100,000 molecules randomly sampled from the ChEMBL training datasets.

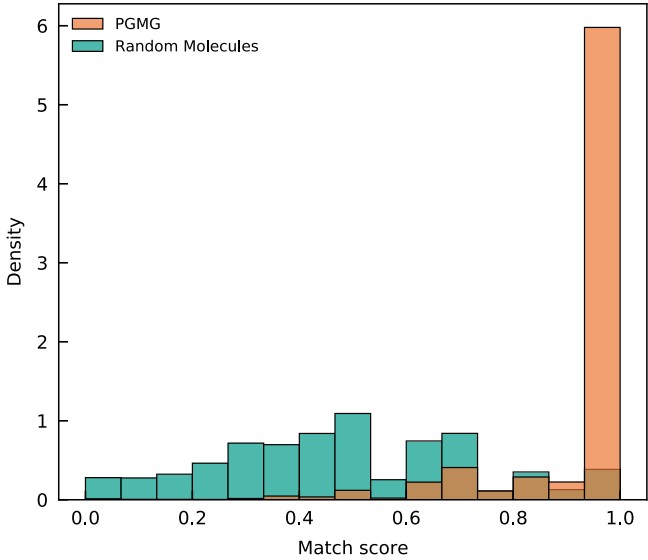

**Fig. 3 | The distributions of the match scores of PGMG-generated molecules compared with randomly selected molecules.** A total of approximately 236,000 molecules were generated using PGMG from random pharmacophore hypotheses extracted from the test dataset and the match scores are calculated and compared with the results of molecules randomly sampled from the training dataset.

pharmacophore-guided docking experiment to assess the binding affinities of the molecules generated by PGMG, which yields similar results. A more detailed descriptions of the pharmacophore-guided docking experiment can be found in the 'Pharmacophore-guided docking' section of the Supplementary Information (Supplementary Fig. 7).

To evaluate whether PGMG can generate drug-like molecules, we further predict the pharmacokinetics properties (absorption, distribution, metabolism, excretion) and toxicity (ADMET) of the top 1000 molecules. As shown in Fig. 4b, most of the molecules generated by PGMG satisfy the pharmacokinetic properties and toxicity constraint for drug candidates, according to the standard proposed by ADMET lab 2.0[36]. A comprehensive comparison of ADMET between the generated molecules and the known bioactivity molecules can be found in Supplementary Fig. 8 and Supplementary Fig. 9.

We have also compared PGMG with other recent methods that can generate bioactive molecules, including RELATION[21], Pocket2Mol[20] and Seq2Seq[15]. We use these methods to generate 10,000 molecules for AKT1 and CDK2 and evaluate them using: (1) the ratio of available molecules, (2) the average SA, (3) the average docking score, (4) the average alignment score between generated molecules and pharmacophore hypotheses (pharmacophore score) and (5) computational time. As shown in Table 2, PGMG achieves the best ratio of available molecules and a top docking score while still maintaining low computational time. We also find that PGMG has a pharmacophore score similar to RELATION, suggesting the consistency between the shortest-path-based pharmacophore and Euclidean distance-based pharmacophore.

## Demonstration of PGMG's application in structure-based drug design

Structure-based drug design (SBDD) utilises the 3D target structure as determined by experimental or homology modelling to design ligands with specific electrostatic and stereochemical features, aiming to achieve a high receptor binding affinity. Here, we consider four targets (VEGFR2, CDK6, TFGB1 and BRD4) with pharmacophore hypotheses collected from the literature[37–41], which are initially built using a ligand-receptor complex as examples to further demonstrate the

performance of PGMG in SBDD. The pharmacophore hypotheses were slightly modified according to the structure of the protein-ligand complex. We compared the top docking score conformations of the generated molecules with the top docking score conformation of the reference ligands in the crystal complex as obtained from AutoDock Vina. As shown in Fig. 5, most of these molecules share interactions with the same amino acid residues as the reference ligands, which indicates that the PGMG-generated molecules are capable of fitting into the binding sites of the reference ligands.

Because of pharmacophore constraints, the generated molecules and reference molecules overlap with the predicted conformations in the pharmacophore region. A greater number of pharmacophore points means a more specific restriction. For example, in Fig. 5a–c, with a pharmacophore restriction that has six pharmacophore points, the generated molecules of VEGFR2 (PDBID: 1YWN) have very similar scaffolds as the reference. As for CDK6 (PDBID: 2EUF) and TGFB1 (PDBID: 6B8Y), despite the structural differences, the generated molecules (Fig. 5e–g, i–k) share common important functional groups as the reference ligands (Fig. 5h, l). On the other hand, the generated molecules shown in Fig. 5m–o possess novel scaffolds of hydrogen bonds with N140. Furthermore, we assess the druggable using SA and hERG, where SA is designed to estimate the SA of molecules, and hERG is the predicted probabilities of hERG inhibition, a toxicity metric to assess the effects of compounds on the heart. These generated molecules perform well on SA and hERG, suggesting that PGMG can design molecules that not only fit well into the binding site, but that also exhibit drug-like quality in the SBDD.

## Demonstration of PGMG's application in ligand-based drug design

Ligand-based drug design is capable of designing drug molecules that are based on the superposition of known active molecules when the target is unknown or binding site is unclear. Here, we consider squalene oxidase, which is the target for ringworm, superficial skin fungal infections and other diseases. Butenafine and terbinafine are typical inhibitors of squalene oxidase[42]. However, these inhibitors are prone to drug resistance and side effects, including skin erythema, burning and itching. Therefore, it is critical to design novel squalene oxidase inhibitors. Here, we have generated 200 molecules using a pharmacophore hypothesis constructed from squalene oxidase inhibitors.

As shown in Fig. 6, these generated molecules align well with the active conformation of terbinafine, which is obtained from molecular dynamics simulation[43]. The listed molecules match well with the desired pharmacophore features, including two hydrophobic groups, a cation and an aromatic ring centre. Furthermore, PGMG captures the equivalence of the different substructures under the same pharmacophore features. For example, it matches the aromatic ring with pyrrole, thiophene and pyrimidine and the hydrophobic group with aliphatic, cycloalkane and benzene. This result shows that PGMG can generate diverse molecules while maintaining the important properties of the substructures that are the same as the known inhibitors.

To further assess the pharmacokinetics and toxicity of the generated molecules, we have calculated the TPSA and SA and predicted the hERG of the generated molecules. TPSA is a molecular descriptor used to measure the polar surface area of a molecule, and it has an application in predicting drug permeability. Of the six molecules generated by PGMG, their TPSA, SA and hERG values are within the rational range. The results suggest that PGMG can generate molecules that match the pharmacophore and meet the overall criteria for TPSA, SA and hERG.

## A showcase of PGMG application in scaffold hopping

Scaffold hopping refers to the acquisition of molecules with novel scaffolds by replacing the chemical core structure while maintaining some essential features of the known active compounds. It has been

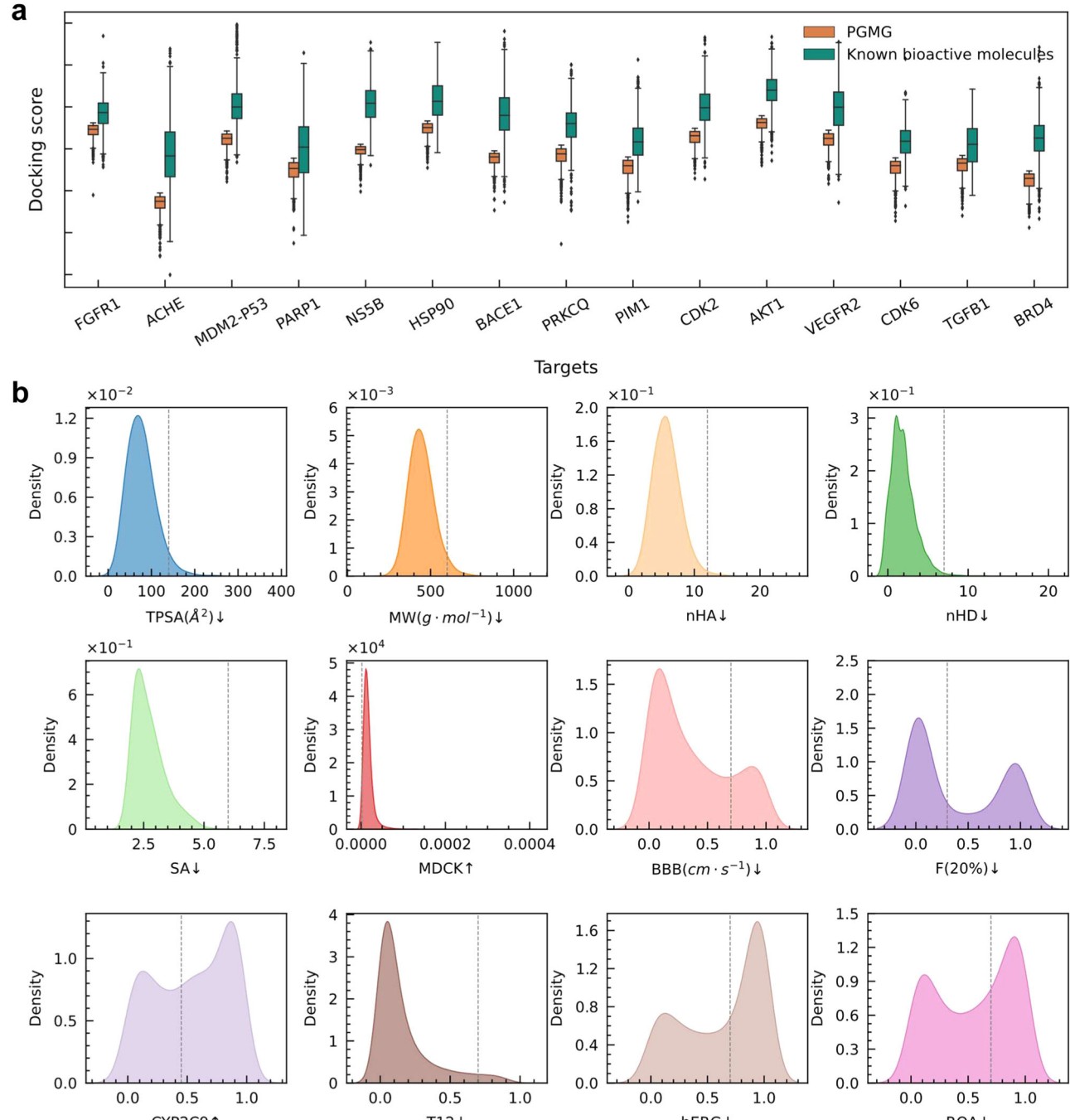

**Fig. 4 | Docking scores and properties distribution of PGMG-generated molecules. a** The distributions of the docking scores of the top 1,000 molecules generated by PGMG over 15 targets compared with those of the top 1000 known bioactive molecules (using a threshold of pChEMBL value >4). The pChEMBL value[58] is the negative logarithm of the molar IC50, EC50, Ki, Kd, or Potency, and it allows these roughly comparable measures to be compared. The median is represented by the centerline of the boxplot, while the first and third quartiles are indicated by the bounds of the box. The whiskers represent the 1.5 interquartile range (IQR). **b** Distributions of the ADMET properties of the top 1000 molecules generated by PGMG. The dashed lines represent the thresholds of these properties, for which an upward arrow indicates that values higher than the threshold are preferred, while a downward arrow indicates that values lower than the threshold are preferred. TPSA represents the topological polar surface area, suitable when: 0–140 (Å²); MW denotes the molecular weight, suitable when: 100–600; nHA represents the

number of hydrogen bond acceptors, suitable when: 0–12; nHD represents the number of hydrogen bond donors, suitable when: 0–7; SA is the synthetic accessibility score, suitable when: <6; the predicted Madin–Darby Canine Kidney cells (MDCK) measures the uptake efficiency of a drug into the body, suitable when: >2 × 10⁻⁶ (cm/s); BBB is the predicted probability of a drug to cross the blood-brain barrier to its molecular targets, qualified value: 0–0.7; F(20%) is the predicted probability of molecules with a human oral bioavailability <20%, suitable when: <0.3; CYP2C9 assesses drug metabolism reactions, and the value is the predicted probability of being an inhibitor; T12 assesses the half-life of the drug, and the value of T12 is the predicted probability of the half-life ≤3; hERG evaluates whether the molecule is toxic to the heart, and the value of hERG is the predicted probability of being inhibiting to the human ether-a-go-go gene; ROA measures acute toxicity in mammals. The value of ROA is the predicted probability of being toxic.

widely applied to generate novel backbones to improve physico-chemical and ADMET properties or to arrive at patentable analogues. As pharmacophores define the chemical features that are essential for biological activity, they can be employed to guide scaffold replacements[44–46]. Here, we show how PGMG can help scaffold hopping using Lavendustin A as a case study. Lavendustin A is an inhibitor

of epidermal growth factor receptor (EGFR), but it is difficult to cross the cell membrane because of its poor lipophilicity. It has been shown that improving the lipophilicity of Lavendustin A can lead to nano-molar levels of IC50 inhibition activity at the cellular level[47]. We con-struct a pharmacophore hypothesis using Pharao[48], and three pharmacophore features are retained by analysing the binding sites of

**Table 2 | The experimental results of PGMG and other methods aimed at generating bioactive molecules**

| Target | Method | Validity↑ | Uniqueness↑ | Novelty↑ | Ratio of Available Molecules↑ | SA±std↓ | Docking Score±std↓ | Pharmacophore Score±std↑ | Time↓ |
|---|---|---|---|---|---|---|---|---|---|
| CDK2 | PGMG | 0.981 | 0.949 | 0.995 | **92.6%** | **2.49 ± 0.42** | **−9.14 ± 0.45** | 0.78 ± 0.06 | 18 s |
| | RELATION_phar[a21] | 0.361 | **1.000** | **1.000** | 36.1% | 2.77 ± 0.54 | −8.67 ± 0.52 | 0.741 ± 0.07 | **5 s** |
| | RELATION_phar-BO_dock[a] | 0.622 | 0.992 | 0.942 | 58.1% | 2.78 ± 0.57 | −8.67 ± 0.53 | 0.743 ± 0.06 | ~60 h |
| | Pocket2Mol[20] | **1.000** | 0.248 | 0.998 | 24.8% | 4.23 ± 1.26 | −9.09 ± 0.98 | - | 1.5 h |
| | Seq2Seq[15] | 0.953 | 0.796 | 0.999 | 75.8% | 2.76 ± 0.43 | −9.09 ± 0.47 | - | 97 s |
| AKT1 | PGMG | 0.996 | 0.848 | 0.993 | **83.9%** | **2.35 ± 0.46** | **−11.17 ± 0.48** | 0.75 ± 0.12 | 15 s |
| | RELATION_phar[a] | 0.332 | **1.000** | **1.000** | 33.2% | 3.13 ± 0.54 | −9.85 ± 0.70 | 0.79 ± 0.14 | **5 s** |
| | RELATION_phar-BO_dock[a] | 0.541 | 0.94 | 0.951 | 48.4% | 3.12 ± 0.55 | −9.83 ± 0.73 | 0.81 ± 0.14 | ~60 h |
| | Pocket2Mol | **1.000** | 0.313 | 0.997 | 31.2% | 4.29 ± 0.95 | −10.48 ± 1.03 | - | 1.9 h |
| | Seq2Seq | 0.945 | 0.67 | 1.00 | 63.3% | 3.18 ± 0.41 | −10.60 ± 0.59 | - | 103 s |

[a]The validity, uniqueness and novelty of the two RELATION methods taken from the original paper. Because different methods have different numbers of available molecules, the synthetic accessibility (SA) score, docking score and pharmacophore score are calculated as the average of the top 2000 molecules sorted based on their docking scores. 'std': the standard deviation. An upward (or downward) arrow next to each metric indicates that higher (or lower) values represent better performance. Best performance among all methods for each metric is shown in bold.

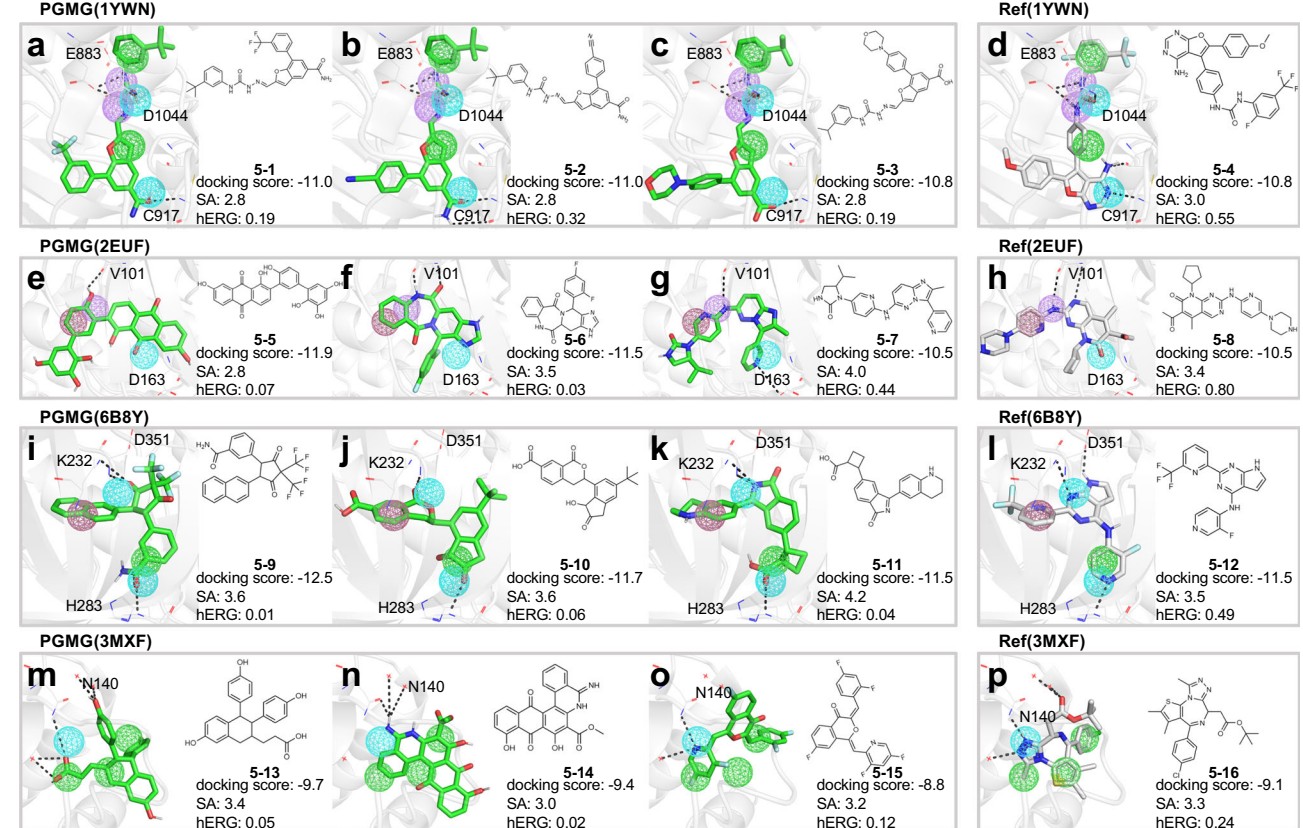

**Fig. 5 | Binding sites of PGMG-generated molecules in a structure-based drug design.** The molecular structure, docking scores, synthetic accessibilities (SA), the predicted probabilities of hERG inhibition (hERG) of the top-ranking molecules and the reference active molecule are given for each target with the corresponding pharmacophore hypothesis: (**a–d**) VEGFR2 (PDBID: 1YWN), (**e–h**) CDK6 (PDBID: 2EUF); (**i–l**) TGFB1 (PDBID: 6B8Y); and (**m–p**) BRD4 (PDBID: 3MXF). Different pharmacophore features are shown in different colours: magenta red (aromatic ring), green (hydrophobic group), purple (hydrogen bond donor), blue (hydrogen bond acceptor). The conformations of generated molecules are acquired through docking.

Lavendustin A in the EGFR protein pocket[49]. Then, we use PGMG to generate molecules for the given pharmacophore hypothesis.

We filter the generated molecules with LogP > 3.41 to obtain molecules with a higher lipophilicity than Lavendustin A. We calculate Tanimoto similarity using Morgan Fingerprints with RDKit[30] between the obtained molecules with three pharmacophore features of the aromatic ring, hydrogen bond donor and hydrophobic group and the EGFR inhibitors acquired from the ExCAPE database[50]. Fig. 7 shows the different scaffolds for the generated molecules with their closest EGFR inhibitors obtained from the ExCAPE database. We have found that some of the generated molecules exhibit high scaffold similarity to the EGFR bioactive molecules in the ExCAPE database, which have not been included in the training set. Furthermore, the generated molecules have the same binding mode as Lavendustin A in EGFR (Supplementary Fig. 10), suggesting that PGMG can discover those inhibitors that have novel scaffolds with only knowledge of Lavendustin A.

## Discussion

In the present study, we have developed a pharmacophore-guided deep learning approach for bioactive molecule generation called

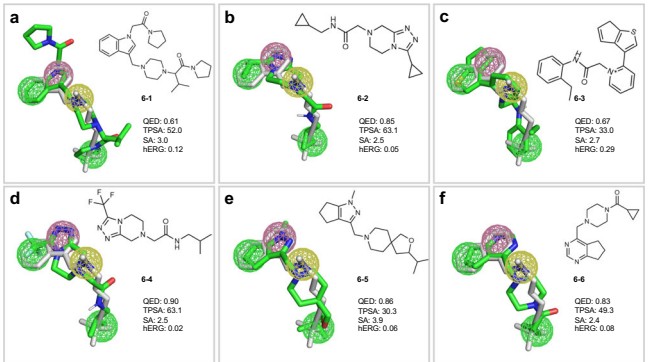

**Fig. 6 | Alignment of terbinafine (grey) and molecules (green) generated by PGMG. a–f** Represent the alignment of six structurally different molecules generated by PGMG with the conformation of terbinafine. The coloured spheres represent different pharmacophore elements, including aromatic ring (red), cation (yellow) and hydrophobic group (green).

PGMG. As the only constraint during the generation process, we use pharmacophores by (1) encoding both the pharmacophore features and spatial information of a given pharmacophore into a complete graph with node and edge attributes and (2) introducing latent variables so that a molecule can be uniquely characterised by a pharmacophore and set of latent variables to handle the many-to-many relationship of pharmacophores and molecules. Our approach offers advantages over current molecule generation methods. First, PGMG provides a way to utilise different types of activity data in a uniform representation, allowing it to overcome the problem of data scarcity. It is also worth mentioning that the training scheme itself does not require any activity data to proceed. Second, pharmacophores incorporate biochemistry knowledge and, thus, are biologically meaningful, hence providing PGMG a strong prior and interpretability into the generation process. Furthermore, a trained PGMG model can be directly applied to different targets without further fine-tuning. We have also developed an easy-to-use web server for PGMG (https://www.csuligroup.com/PGMG) that allows users to generate molecules for any given pharmacophore hypothesis.

We show that PGMG is competent in generating a large number of molecules with docking scores similar to or even better than the known active molecules obtained from the ChEMBL database. With the pharmacophore for certain targets, PGMG can also be utilised to design dual or multitarget molecules. In addition, we expect that PGMG can be adopted to prepare chemical libraries to improve virtual screen efficiency because this can provide a certain number of candidate drug-like molecules for a specified target. The structure-based and ligand-based case study shows that PGMG can generate high-quality bioactivity molecules that match the pharmacophore hypothesis with structural diversity, suggesting that PGMG can be applied to multiple drug design scenarios, such as researching alternative medicine and drug resistance. Finally, the showcase of scaffold hopping demonstrates that PGMG can discover active molecules with novel scaffolds.

De novo drug design is a complicated and situation-specific problem, and computational methods should benefit from the input of prior biochemistry knowledge. PGMG benefits from this idea by leaving the construction of pharmacophore hypotheses to the user. There are multiple ways to form a pharmacophore hypothesis; various information can be used, and different adjustments can be made to enhance the hypothesis. A more accurate hypothesis can be used to

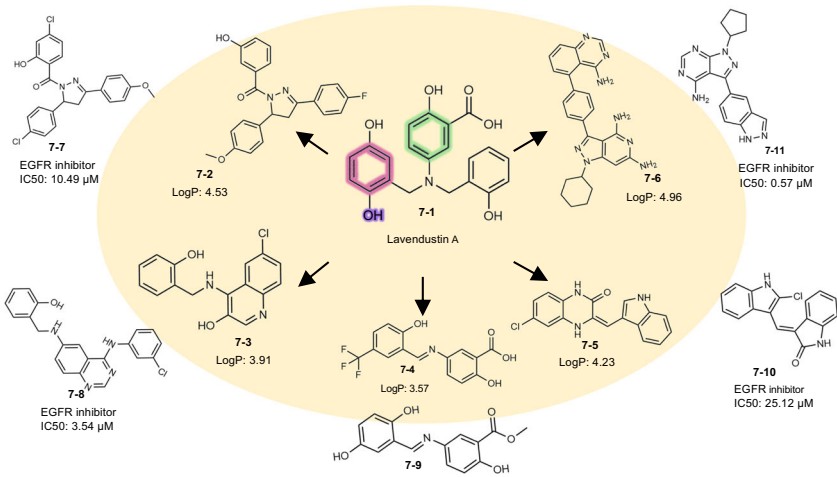

**Fig. 7 | The molecule generated by PGMG with known inhibitors in the case of scaffold hopping.** Molecules generated by PGMG are shown inside the circle, and their closest active nearest neighbours are shown outside the circle. The colours

indicate the pharmacophore features extracted from Lavendustin A: aromatic ring (red), hydrogen bond acceptor (blue) and hydrophobic group (green).

import the quality of the generated molecules. For example, QSAR studies can be used to adjust the hypothesis, which may result in better constraints of the generated molecules. On the other hand, the limitations of PGMG should be acknowledged. For example, PGMG currently does not support exclusion volume in pharmacophore hypotheses. Since we focus on the task of generating molecules with the desired activities, PGMG does not explicitly constrain the properties of the generated molecules. A future direction of our work is to include the exclusion volume and other features in PGMG, making the generated molecules more controllable and malleable. Another direction is to incorporate QSAR analysis into the generative models, which may provide a stronger baseline of pharmacophore hypothesis with enhanced interpretability.

## Methods

### Datasets
We used the ChEMBL 24 dataset containing more than 1.25 million molecules to train PGMG. ChEMBL is a collection of bioactivity data for various targets and compounds from the literature. It contains 13 types of atoms ($T = 13$): H, B, C, N, O, F, Si, P, S, Cl, Se, Br and I. Each bond is either a no-bond, single, double, triple or aromatic bond ($R = 5$).

We also used the ZINC[51] molecule dataset from JTVAE[52] for our ablation study. It contains 220,000 molecules in the training data and 11 types of atoms ($T = 11$): H, B, C, N, O, F, P, S, Cl, Br and I. Each bond is either a no-bond, single, double, triple or aromatic bond ($R = 5$).

The structure of the targets FGFR1 (PDBID: 2FGI), ACHE (PDBID: 4EY7), MDM2-P53 (PDBID: 3JZK), PARP1 (PDBID: 6I8M), NS5B (PDBID: 3PHE), HSP90 (PDBID: 3HHU), BACE1 (PDBID: 2IRZ), PRKCQ (PDBID: 1XJD), PIM1 (PDBID: 3BGQ), CDK2 (PDBID: 4KD1), BRD4 (PDBID: 3MXF), VEGFR2 (PDBID: 1YWN), CDK6 (PDBID: 2EUF), TGFB1 (PDBID: 6B8Y) and AKT1 (PDBID: 4GV1) are downloaded from the PDB[53] database.

### Representation of pharmacophores and molecules
A pharmacophore hypothesis consists of several chemical features and their spatial descriptions and is represented by a fully connected graph with chemical feature types as node attributes and distances as edge weights (a detailed description of the pharmacophore graph and preparation of the pharmacophore graph is included in the Supplementary Information). We have applied a state-of-the-art graph neural network, Gated-GCN[28], to embed the graph by considering node attributes and edge attributes.

Molecules are represented in SMILES format. Symbols of stereochemistry like '@' '/' are removed because stereochemistry information does not exist in the graph representation of a pharmacophore, and it is not difficult to list all the stereoisomers of a molecule. Then, the SMILES string is separated into a sequence of tokens corresponding to heavy atoms and structural punctuation marks. For example, the SMILES string 'C(C[NH2-])OC( = O)Cl' will be split into 'C/(/C/[NH2-]/)/O/C/( = /O/)/Cl', where each token will be embedded into a vector.

### Encoder and decoder
An illustration of the encoder and decoder networks can be found in Fig. 1. The encoder and decoder network have been adapted from the standard transformer[29] architecture, with each consisting of several layers of stacked transformer encoder and transformer decoder blocks. The difference between the transformer encoder and decoder blocks is that the encoder block uses only self-attention modules and the decoder block uses cross-attention modules to incorporate the context in the generation process. Some modifications have been made to handle our inputs and better suit the variational autoencoder structure of PGMG.

We first calculate the latent variables $z$ of molecule $x$ given pharmacophore $c$ by the encoder network. The encoder input is a concatenation of molecule and pharmacophore features. Following BART[31], positional and segment encoding are added to the following input sequence:

$$\text{Input}_{encoder} = \left( E'_p; E'_m \right) \tag{2}$$

$$E'_{m_i} = E_{m_i} + SE_m + PE_i \tag{3}$$

$$E'_{p_j} = E_{p_j} + SE_p \tag{4}$$

where $Input_{encoder}$ is the input representation, $E_{p_j}$ is the j-th pharmacophore feature vector, $E_{m_i}$ is the i-th token embedding of molecule features, $SE_m$ and $SE_p$ are two segment embedding vectors for molecule features and pharmacophore features, and $PE_i$ is the positional embedding for the i-th token. After several layers of transformer encoder block, the molecule features are averaged by an attention pooling layer to obtain the final molecule representation $h_x$. $h_x$ is then fed into two separate subnetworks to compute the mean $\mu$ and log variance $\log \Sigma$ of the posterior variational approximation. Latent variables $z$ are then sampled from the normal distribution $N(\mu, \Sigma)$.

The decoder network takes the latent variables $z$ and pharmacophore features as the input:

$$\text{Input}_{decoder} = \left( E'_p; E'_z \right) \tag{5}$$

$$E'_{z_i} = z_i + SE_z + PE_i \tag{6}$$

where $E'_p$ is calculated using Eq. (4), $SE_z$ is the segment embedding for the latent variables, and $PE_i$ is the positional embedding for i-th token. The decoder then uses $input_{decoder}$ to generate target SMILES autoregressively. Each token is determined based on previously generated tokens and context:

$$o_i = \arg\max_{o_i} P\left(o_i | o_{<i}, c, z\right) \tag{7}$$

where $o_i$ is i-th generated token.

### Loss function
PGMG's model is trained in an end-to-end manner. The loss function consists of three different terms: KL loss, language modelling loss (LM loss) and mapping loss. The first two terms are derived from the evidence lower bound (ELBO) of the log-likelihood $\log P_\theta(x|c)$:

$$\log P_\theta\left(x | c_p\right) = \log \int P_\theta(x|c, z) P_\phi(z|c) dz \geq - \text{KL}(P_\phi(z|x, c) || P_\theta(z|c)) + E_{P_\phi(z|x, c)} \left[ \log P_\theta(x|z, c) \right] \tag{8}$$

where KL denotes the Kullback–Leibler divergence and where we assume $P_\theta(z|c)$ the prior distribution of $z$ to be a standard Gaussian $N(0, I)$. We call $\text{KL}(P_\phi(z, |, x, c) || P_\theta(z, |, c))$ KL loss, and it serves as a regulation term to mitigate the gap between the true prior distribution of $z$ and the posterior distribution while making the latent space of $z$ smoother. The expectation term $E_{P_\phi(z, |, x, c)} \left[ \log P_\theta(x, |, z, c) \right]$ is estimated through Monte Carlo estimation with one data point for each sample[54]. Because $x$ takes the form of the SMILES string, we refer to it as the LM loss.

The third part of PGMG's loss function is mapping loss. It evaluates the model's performance in predicting the mapping between heavy atoms and pharmacophore elements. We use mapping loss as a regulation term to help alleviate the problem of posterior collapse. The mapping score of the i-th pharmacophore $p_i$ and j-th output token $o_j$ is

calculated as follows:

$$s_{mapping\,ij} = \sigma\left(g\left(W_p E_{p_i}\right) \odot g\left(W_o E_{o_j}\right)\right) \qquad (9)$$

where $s_{mapping\,ij}$ is the mapping score, $E_{p_i}$ and $E_{o_j}$ are the embedding vectors of $p_i$ and $o_j$, respectively, $W_p$ and $W_o$ are two learnable matrices to project two different embeddings into the same space, $\odot$ is the dot product, $\sigma$ is the sigmoid function and $g$ is the ReLU function. The calculation of the mapping scores can be vectorised as follows

$$s_{mapping} = \sigma\left(g\left(W_p E_p\right) g\left(W_o E_o\right)\right) \qquad (10)$$

Since the SMILES format contains tokens other than atom symbols, we mask them when calculating the mapping loss. The mapping loss is then calculated as the cross-entropy of the masked scores and labels. An illustration of the masked mapping score and label is given in Supplementary Fig. 11.

### Training details and model parameter settings

During training, we inject noise into the input to make the training more robust by using the infilling scheme. Some random subsequences in every input sequence are replaced by a single [mask] token. The teacher forcing technique is applied to the generation process during training, by which we replace the previously generated tokens with the ground truth to produce the next token. Aside from the mapping loss introduced before, another approach we use to alleviate posterior collapse is KL annealing[55], where an increasing coefficient is used to control the size of KL loss.

We use the same model parameters in both the ChEMBL and ZINC datasets. The hidden dimension is 384. The transformer encoder blocks and transformer decoder blocks are stacked eight times. We use an eight-head attention, and the feed-forward dimension is 1024. We use an Adam optimiser to train the model with a 3e−4 learning rate and a 1e−6 weight decay rate. Cosine learning rate annealing is applied with a cycle length of four epochs. We use the gradient clipping technique and set the maximum gradient as five. Since the ChEMBL dataset contains many more molecules compared with the ZINC dataset, it requires fewer training epochs to reach a similar validation performance. Thus, the number of training epochs for the former is 32 and 48 for the latter.

### Evaluation

Four different metrics, including validity, uniqueness, novelty and ratio of available molecules, are employed to evaluate the ability to generate novel molecules. Validity is the percentage of chemically valid molecules with the generated SMILES. Uniqueness measures how many valid molecules are nonrepetitive. Novelty refers to the percentage of chemically valid molecules not generated in the training set. The ratio of available molecules is the proportion of novel molecules in all generated results. These metrics are calculated as follows:

$$validity = \frac{\#\,of\ chemically\ valid\ SMILES\ (\#\,of\ molecules)}{\#\,of\ generated\ SMILES} \qquad (11)$$

$$uniqueness = \frac{\#\,of\ unique\ molecules}{\#\,of\ molecules} \qquad (12)$$

$$novelty = \frac{\#\,of\ novel\ molcules}{\#\,of\ unique\ molecules} \qquad (13)$$

$$ratio\ of\ available\ molecules = \frac{\#\,of\ novel\ molecules}{\#\,of\ generated\ SMILES} \qquad (14)$$

We use the match score to indicate the match degree of the generated molecules to the specified pharmacophore (see the calculation of match score section of the Supplementary Information for details).

The docking score of AutoDock Vina[35] is used as a proxy for the binding activity of the generated molecules to the target[35]. We use AutoDock Vina to perform semiflexible docking with the default parameter, where the flexibility of ligands is considered to dock into a rigid receptor. The central coordinates of the box are calculated as the average coordinates of each heavy atom in the ligand. The size of the box is determined by the size of the ligand in the PDB complex. The analysis shows that water in the BRD4 (3MXF) pocket affects receptor binding to the ligand, so we conduct hydrated docking for 3MXF[56].

We also use ADMETlab 2.0[36] to predict the ADMET properties of the generated molecules and assess the drug-like potential of the generated molecules.

The pharmacophore score in Table 2 is calculated with Align-it[48], a tool to align molecules with the pharmacophores hypotheses:

$$pharmacophore\ score = \frac{V_{overlap}}{V_{ref}} \qquad (15)$$

where $V_{overlap}$ is the overlapping volume between the given pharmacophore elements and the reference and where $V_{ref}$ is the volume of the reference pharmacophore elements[36] to predict the ADMET properties of the generated molecules and assess the drug-like potential of the generated molecules. The computational time is calculated using a NVIDIA Tesla V100s GPU card, except for RELATION$_{phar}$-BO$_{dock}$, of which the computational time is obtained from the original paper. For Pocket2Mol[20] and Seq2Seq[15], we run the experiment using the code and model weights provided by the authors. However, for Pocket2Mol, we make a small modification in its code to disable it from filtering the duplicated results and outputting only unique molecules. As for RELATION[21], the results are acquired using ReMODE[57], a web server developed by the authors of RELATION. Since the web server will automatically filter the generated molecules, validity, uniqueness and novelty are obtained from the original paper of RELATION. The pharmacophore hypotheses used for fine-tuning RELATION are provided by the author and later used in the generation process of PGMG.

We also conducted an ablation study to see the performance of variants of PGMG. The details can be found in the 'Ablation Study' section of the Supplementary Information (Supplementary Table 3).

### Reporting summary

Further information on research design is available in the Nature Portfolio Reporting Summary linked to this article.

## Data availability

The dataset used to train and evaluate the model is provided at https://github.com/CSUBioGroup/PGMG. Other data including generated molecules are provided in Supplementary Data files 1–3. Source Data for Figs. 2–4, Tables 1, 2, and Supplementary Figs. 1–9 are provided as a Source Data file. Source data are provided in this paper.

## Code availability

The source code is available at https://github.com/CSUBioGroup/PGMG, which has also been deposited in the Zenodo under accession code https://doi.org/10.5281/zenodo.8195825.

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

## Acknowledgements

This work is financially supported by the National Natural Science Foundation of China under Grants (No. 61832019 to M.L.), Hunan Provincial Science and Technology Programme (2019CB1007 and 2021RC4008) [M.L.], Academy of Finland (No. 317680 to J.T.), and European Research Council (No. 716063 to J.T.).

## Author contributions

M.L. and J.T. supervised the study. H.Z., R.Z. and M.L conceived the initial idea. H.Z. collected and preprocessed the data and R.Z. designed the model. R.Z. performed the generation experiments and H.Z. performed the case studies. D.C. provided the Molecular Operating Environment (MOE) for pharmacophore-guided docking experiments. H.Z., R.Z., J.T. and M.L. wrote the paper.

## Competing interests

The authors declare no competing interests.
