## [Peer review file · Nature Communications]

REVIEWER COMMENTS

Reviewer #1 (Remarks to the Author):

In this paper, Zhu et al. developed a novel pharmacophore-guided deep learning approach called PGMG for the bioactive molecular generation. They used a GatedGCN to encode pharmacophore features and the pharmacophore embedding served as inputs for the conditional generative model. In this way, the model was able to generate novel compounds by satisfying the pharmacophore constraints given as latent variables. I found the paper interesting to read, presented with high technicality and novelty and demonstrated using the latest deep learning approaches. One of the biggest strengths of the program is that the pharmacophore constraints/conditions were integrated as part of the deep learning module without the need of additional post-filtering steps. Below are a few of my comments:

1. My major concern is that the pharmacophore is encoded in 2D while the compound is encoded in 1D/2D and therefore I suspect that the generated molecules from the model were conformed to the 2D pharmacophore constraints not 3D. However, the authors seem to claim that the generated molecule could bind to 3D structural targets with 3D pharmacophore constraints, at least in one-shot, as demonstrated in the "structure-based drug design" section. I'm not convinced this is the case unless the molecule in consideration is flat or with the help of further structure-based screening.
2. The "structure-based drug design" section needs additional details following my point 1 above. How was the pharmacophore hypothesis developed? How were the compound conformations generated and were the lowest-energy conformation picked (noted that there could be multiple ligand conformations/potential pharmacophore combinations generated during the structural enumeration step)? At the minimum, it would be great to highlight the pharmacophores within the structure model in figure 4 to demonstrate the points and for visualization.
3. There is some attempt to demonstrate correlations between 2D vs 3D Euclidean distance pharmacophore in the supplementary information. While the authors show that there is a strong pair-wise correlation, a pharmacophore hypothesis usually contains >3 pharmacophore points and here the comparisons from 2D to 3D mapping were pair-wise making it sort of trivial. I would suggest comparing the correlation of all the pharmacophore points together to demonstrate that the original pharmacophores were maintained.
4. Although it may not be necessary, it could further strengthen the impact of the paper if the authors could test 1-2 predicted compounds for experimental target binding from figure 3b.

6. The authors performed "in-silico binding" of predicted compounds using autodock vina. However, it is unclear from the supplementary information if the results in figure 3b were based on rigid or flexible docking. If it is the former, it is best to specify given that by default the program performs flexible docking.

7. Another major limitation of predicted molecules from the current generative models is synthetic feasibility, particularly of those SMILES-based generative models. In addition to the several metrics used to access the performance in the evaluation, I would also recommend evaluating the synthetic accessibility scores of the predicted compounds. See, for example, <https://jcheminf.biomedcentral.com/articles/10.1186/1758-2946-1-8>

8. The authors make some claims that the program can predict novel compounds without/or with limited structural activity data. I think this is an advantage of the ligand-based drug design approach in general but not specific to the PGMG program. While it is impressive that the PGMG program can generate potentially bioactive compounds satisfying any pharmacophore constraints, noted that not all pharmacophore points are essential for binding and will still require extensive QSAR study to validate a pharmacophore hypothesis.

Reviewer #2 (Remarks to the Author):

PGMG: A Pharmacophore-Guided Deep Learning Approach for Bioactive Molecular Generation (Zhu et al.)

The authors propose a method to condition SMILES generation with pharmacophores. They follow traditional VAE methods to embed SMILES in a latent space using Transformer encoders/decoders, and pre-process pharmacophores as fully connected graphs between the spatial features into a conditioning vector that will be concatenated to the SMILES embedding for encoding or latent space vector for decoding. They show that their method reaches satisfactory validity, uniqueness and novelty of generated molecules compared to other popular SMILES generative models, while reaching high matching to conditioning pharmacophore. They further demonstrate that generated molecules using structure-based pharmacophore have on average better binding affinity as estimated by Vina, while being in acceptable ranges of predicted ADMET properties. These putative high affinity molecules tend to adopt similar binding modes to active molecules or find new interactions. They also show examples of ligand-based pharmacophore matching for terbinafine. Finally, they use the model to generate compounds for an EGFR modified pharmacophore to increase logP of the initial active Lavendustin A, and show that they retrieve molecules similar to other actives that were not included in training.

Overall this is interesting work that, due to the focus of using pharmacophores for design, is also largely novel (though one key comparison to a related method is missing which is close, see below).

Major items:

- This work does not compare nor cite the RELATION work that also used pharmacophores as input for the model: <https://pubs.acs.org/doi/full/10.1021/acs.jmedchem.2c00732>. Can a common benchmark be used to establish relative performance of methods?

- As is often the case in the area there is no experimental validation of the generated molecules – this would greatly increase the practical relevance of the work (I leave it up to the editor to judge whether purely computational approaches and validation are of sufficient interest to the journal, this is a frequent difference between computer science and life science thinking)

- This is also reflected in the wording – the authors need to make clear that properties are predicted, interactions putative (if based on docking), etc. E.g. “To further assess the pharmacokinetics and toxicity of the generated molecules, we calculate the TSPA,280

SA, and hERG of the generated molecules.” (line 280) – PK and toxicity get assessed by in vivo experiments, not by ‘calculating TSPA (TPSA?), SA and hERG’ activity (etc, throughout text)

- The shortest-path proxy sounds great, but are there any cases where it fails? For example, in the scatterplot Figure S1, we can see points with low Euclidean distance (< 4) and high shortest-path distance (>8), likely to correspond to “folded” conformations. I would expect the model to “succeed” in having this shortest-path matching, but failing at the 3D Euclidean distance matching. Also, 1000 molecules for Figure S1 seems a bit low to evaluate the rate of the distance mismatches.

- In the ablation study, what exactly is random sampling? Is it uniform sampling or something else?

- The training data uses random pharmacophores extracted from molecules. Does it include active compounds towards the target used in the structure-based pharmacophore? For instance, Figures 4a to c shows very similar scaffolds, is there a way that the model has seen a compound that shows a more or less specific pharmacophore of VEGFR2 before?

- In the structure-based pharmacophore generation, the comparison is made between the top-1000 generated molecule and a variable number of known actives for the target. It would maybe be fair to compare with the top-1000 known actives (given that some low-activity compounds can be a result of exploring SAR, and not just optimization towards higher activity).

- On the ligand-based pharmacophore for terbinafine, the authors mention that they obtained the active conformation of terbinafine from Drugbank. Is it really the bioactive conformation (from experimental determination), or is it a low-energy conformation? If bioactive, then a comparison could be made with the determined target, if not, then the extracted pharmacophore might not reflect the one of the bound pose.

- On the modified pharmacophore of Lavendustin A, the authors only test among the compounds with perfectly matching pharmacophore the logP and similarity to known actives, which are convincing

candidates. Another validation could be docking to known EGFR structure to try and evaluate if the binding mode is similar.

- A missing measure in this work (and lots of other generative modelling research in general) is the generation time. For instance, PGMG is compared to popular SMILES generator, reaching satisfactory metrics. However, if current work generates 1000 molecules per second while the VAE approach generated 10000 molecules, you will obtain a higher number of valid, unique and novel (that the authors refer to as available) molecules in the same amount of time.

- How does the current work compare with SMILES generator conditioned on docking score. I would expect PGMG to be faster on the generation, but I wonder what the difference between the predicted binding affinity range and match score range would be.

- As you mention, including exclusion volume in pharmacophore is a relevant perspective. It would be great to see some examples where those could be efficient, as I expect the current approach to maximize the match score but at the cost of not being able to bind in the real pocket without clashes (cf. remark on docking compounds for EGFR pharmacophore)

Minor comments:

I.10: bioactive instead of bioactivate

I.120: pharmacophore instead of pharamacophore

I.214: calculates seems optimistic, Autodock vina only predicts/estimates the binding affinity

I.238: could be rephrased: "share interaction with the same amino acid residues"

I.280 and I.282: TPSA and not TSPA

I.284: Figure 5 should be bold font to match previous formatting

I.362: Would be easier for the reader to directly have the PDB ids in the methods, to avoid going back to the results section

I.452: novel molecules instead of novelty molcules

Detailed comments on the supplementary information:

I.9: Basefeatures.fdef and not Basefeatues.fdef. Might be great to add a direct link as well from the official rdkit github repo to know where to find it

I.16: feature instead of features

I.42: a node set instead of anode set

Reviewer #3 (Remarks to the Author):

This paper presents a deep-learning based approach to generate molecules with matched pharmacophore structures. Technically, the method appears to be interesting and has its own value, there are serious concerns about the evaluation of the method. Thus, I'm not fully convinced that the paper is novel and efficient enough to be published in Nature Communications.

1. The ratios of valid and novel compounds are compared with ORGAN, VAE, SMILES LSTM, and Syntalinker. Most compared methods are published a while ago and many more efficient methods are introduced. The comparison with relatively early models is not fully convincing.

2. The improvements in binding affinities, based on the Autodock-vina score, are tested with only four targets. It is not clear whether this improvement is general and transferrable to other targets. More comprehensive tests are necessary.

3. The result shown in Figure 3a is the only result that shows how much generated molecules are consistent with a given pharmacophore restraint. A comparison with random molecules appears to be obvious. More comparisons with existing molecular generation methods that use pharmacophore information are needed.

4. The lead optimization results shown in Figure 6 do not fully support whether this method is efficient for lead optimization. The generated molecules show large variations from the seed molecule, which appears to be too large for lead optimization. Also, the argument, "Based on the assumption that structurally similar molecules have similar properties, the similarity result demonstrates that molecules generated by PGMG have a probability of inhibiting EGFR." appears to be weak.

Response letter

On behalf of all the contributing authors, we would like to express our sincere appreciation for the reviewers' constructive comments and helpful suggestions regarding our article entitled '*PGMG: A Pharmacophore-Guided Deep Learning Approach for Bioactive Molecule Generation*'. In response to the feedback from the reviewers, we have made substantial revisions to our manuscript and have added extra materials to strengthen our results. All the changes have been highlighted in the revised manuscript. We hope that this response letter resolves the reviewers' concerns.

Reviewer #1:

In this paper, Zhu et al. developed a novel pharmacophore-guided deep learning approach called PGMG for the bioactive molecular generation. They used a GatedGCN to encode pharmacophore features and the pharmacophore embedding served as inputs for the conditional generative model. In this way, the model was able to generate novel compounds by satisfying the pharmacophore constraints given as latent variables. I found the paper interesting to read, presented with high technicality and novelty and demonstrated using the latest deep learning approaches. One of the biggest strengths of the program is that the pharmacophore constraints/conditions were integrated as part of the deep learning module without the need of additional post-filtering steps. Below are a few of my comments:

Authors' response:

We thank the reviewer for the overall positive comments. Please find the point-by-point responses below.

1. My major concern is that the pharmacophore is encoded in 2D while the compound is encoded in 1D/2D and therefore I suspect that the generated molecules from the model were conformed to the 2D pharmacophore constraints not 3D. However, the authors seem to claim that the generated molecule could bind to 3D structural targets with 3D pharmacophore constraints, at least in one-shot, as demonstrated in the "structure-based drug design" section. I'm not convinced this is the case unless the molecule in consideration is flat or with the help of further structure-based screening.

Authors' response:

It is true that the shortest-path-distance-based pharmacophore constraint is a 2D constraint. However, we show that the generated molecules that match a given 2D pharmacophore can bind to 3D structural targets.

Previous studies have explored using 2D pharmacophore in the prediction of active molecules and Structure-Activity Relationships (SAR) analysis¹⁻³, suggesting that the 2D pharmacophore contains sufficient information for identifying active molecules. Metivier et al². used 2D pharmacophore-based networks to study the SAR of different active molecules for a specific target and identify active molecules for different targets. Nakano et al³. proposed a NScaffold method to rank 2D pharmacophore hypotheses and identified hypotheses that could successfully retrieve known important protein-ligand interactions.

In our study, we have conducted two experiments to show how the shortest-path-based pharmacophore can be used to generate novel active-like molecules. In the first experiment, the ability of the generated molecules to bind to the 3D structure of targets was tested using the AutoDock Vina⁴ docking algorithm. We have expanded the scale of this experiment in the revised manuscript from 4 to 15 targets, whose pharmacophore hypotheses were obtained from the literature. We used PGMG to generate 10,000 molecules for each pharmacophore hypothesis and compared the docking scores of the top 1,000 generated molecules and top 1,000 experimentally validated molecules collected from the ChEMBL database. As shown in **Figure R1**, the docking scores for the two sets of molecules are similar, suggesting that these PGMG-generated molecules can bind to the 3D structure of the targets.

Figure R1 (Figure 4a). Box plot of the docking scores of the top 1,000 molecules generated by PGMG and top 1,000 experimentally validated molecules for 15 targets.

Furthermore, we have added new comparison results in the revised manuscript (**Table R1**), in which the PGMG-generated molecules show stronger docking affinities than those molecules generated by methods that directly use the 3D receptor structure as an input (RELATION⁵ and Pocket2Mol⁶). Therefore, this result supports the idea that the shortest-path-based constraint can be used to generate active-like molecules. The detailed description of this new experiment has been updated in the ‘Results’ section of the revised manuscript.

Table R1. Performance of PGMG compared with other methods for generating bioactive molecules.

Target	Method	Validity ↑	Uniqueness ↑	Novelty ↑	Ratio of available molecules ↑	SA↓	Docking Score↓	Pharmacophore Score↑	Time↓
CDK2	PGMG	0.981	0.949	0.995	92.6%	2.66	-8.68	0.782	18s
	RELATION _{phar} *	0.361	1	1	36.1%	2.86	-8.32	0.742	5s
	RELATION _{phar} -BO _{dock} *	0.622	0.992	0.942	58.1%	2.87	-8.30	0.742	~60h
	Pocket2Mol	1	0.248	0.998	24.8%	4.28	-8.38	-	1.5h
	Seq2Seq ⁷	0.953	0.796	0.999	75.8%	2.87	-8.69	-	97s
AKT1	PGMG	0.996	0.848	0.993	83.9%	2.45	-10.84	0.762	15s
	RELATION _{phar} *	0.332	1	1	33.2%	3.11	-9.85	0.790	5s
	RELATION _{phar} -BO _{dock} *	0.541	0.94	0.951	48.4%	3.10	-9.83	0.813	~60h
	Pocket2Mol	1	0.313	0.997	31.2%	4.25	-10.40	-	1.9h
	Seq2Seq	0.945	0.67	1	63.3%	3.20	-10.43	-	103s

*The validity, uniqueness and novelty of the two RELATION methods are taken from the original paper. Because different methods have different numbers of available molecules, the synthetic accessibility scores (SA), docking score and pharmacophore score are calculated as the average of the top 2,000 molecules sorted based on their docking scores.

On the other hand, as shown in **Figure R2**, the generated molecules are also likely to have a conformation that has a high degree of overlap with the 3D pharmacophore hypothesis. Moreover, these generated molecules, which do not match the Euclidean distance based pharmacophore, can also have a strong binding affinity. For example, most of the generated molecules shown in the section titled ‘Demonstration of PGMG’s application in structure-based drug design’ (which are selected among the generated molecules with the best docking scores) show a certain overlap with the original 3D pharmacophore hypotheses despite imperfect match (**Figure R2**). There are some inconsistencies in the positions of the pharmacophore points, but these molecules have predicted the binding sites and affinities similar to those of the reference ligands. In **Figure R2**, the generated molecules of 2EUF (e-g) do not align well with the aromatic ring (red sphere) and hydrogen bond donor (purple) of the given 3D pharmacophore hypothesis, but these molecules still have good docking scores.

Figure R2 (Figure 5). Binding sites of PGMG-generated molecules in a structure-based drug design. The binding sites and pharmacophore hypothesis for the top-ranking molecules are highlighted for four targets: VEGFR2 (1YWN), CDK6 (2EUF), TGF β 1 (6B8Y) and BRD4 (3MXF). Different pharmacophore features are shown, including aromatic ring (red), hydrophobic group (green), hydrogen bond donor (purple) and hydrogen bond acceptor (blue).

As for the molecules displayed in the section titled ‘Demonstration of PGMG’s application in structure-based drug design’, they have been selected among the generated molecules with the highest docking scores, and the bioactive conformation is acquired using the docking programme AutoDock Vina⁴. We have updated the expressions in the revised manuscript to make them clearer.

Taken together, we added more results to show that the shortest-path-based pharmacophore can be used to generate molecules with strong docking scores for different targets and that the performance of PGMG exceeds state-of-the-art methods that directly use the 3D receptor structure as input. These results suggest that the shortest-path-based pharmacophore constraint used in PGMG contains sufficient information to guide the generation of active-like molecules.

Reference

- Horvath D. Topological Pharmacophores. In: *Cheminformatics Approaches to Virtual Screening*. RCS Publishing: Cambridge, UK pp 44–75 (2008).
- Métivier J-P, Cuissart B, Bureau R, Lepailleur A. The pharmacophore network: a computational method for exploring structure–activity relationships from a large chemical data set. *Journal of Medicinal Chemistry* **61**, 3551-3564 (2018).
- Nakano H, Miyao T, Funatsu K. Exploring topological pharmacophore graphs for scaffold hopping. *Journal of Chemical Information and Modeling* **60**, 2073-2081 (2020).
- Trott O, Olson AJ. AutoDock Vina: improving the speed and accuracy of docking with a new scoring function, efficient optimization, and multithreading. *Journal of computational chemistry* **31**, 455-461 (2010).
- Wang M, *et al.* RELATION: a deep generative model for structure-based de novo drug design. *Journal of Medicinal Chemistry* **65**, 9478-9492 (2022).
- Peng X, Luo S, Guan J, Xie Q, Peng J, Ma J. Pocket2mol: Efficient molecular sampling based on 3d protein pockets. In: *International Conference on Machine Learning*. PMLR (2022).
- Uludođan G, Ozkirimli E, Ulgen KO, Karalı N, Özgür A. Exploiting pretrained biochemical language models for targeted drug design. *Bioinformatics* **38**, ii155-ii161 (2022).

2. The "structure-based drug design" section needs additional details following my point 1 above. How was the pharmacophore hypothesis developed? How were the compound conformations generated and were the lowest-energy conformation picked (noted that there could be multiple ligand conformations/potential pharmacophore combinations generated during the structural enumeration step)? At the minimum, it would be great to highlight the pharmacophores within the structure model in figure 4 to demonstrate the points and for visualization.

Authors' response:

In the section titled 'Structure-based drug design (renamed to Demonstration of PGMG's application in structure-based drug design)', the pharmacophore hypotheses were collected from the literature¹⁻⁵ and were initially built according to the receptor-ligand complex structures.

The displayed conformations were computationally determined through docking by AutoDock Vina with the best docking scores. We have added a description of the docking parameters in the 'Methods' section in the revised manuscript.

Following your suggestions, we have highlighted the pharmacophores in Figure 4 (renumbered to Figure 5 in the revised manuscript), as shown as **Figure R2** in the response letter. It can be seen that, in **Figure R2 (a-c)**, the generated molecules with a pharmacophore constraint of six points display similar scaffolds as the reference. The positions of the pharmacophore sites in **Figure R2 (d-p)** do not match 100% with those of the reference ligands, but they have predicted binding sites and affinities similar to those of the reference ligands.

Reference

1. Lee K, *et al.* Pharmacophore modeling and virtual screening studies for new VEGFR-2 kinase inhibitors. *European journal of medicinal chemistry* **45**, 5420-5427 (2010).
2. Shawky AM, Ibrahim NA, Abourehab MA, Abdalla AN, Gouda AM. Pharmacophore-based virtual screening, synthesis, biological evaluation, and molecular docking study of novel pyrrolizines bearing urea/thiourea moieties with potential cytotoxicity and CDK inhibitory activities. *Journal of enzyme inhibition and medicinal chemistry* **36**, 15-33 (2021).
3. Roskoski Jr R. Cyclin-dependent protein serine/threonine kinase inhibitors as anticancer drugs. *Pharmacological research* **139**, 471-488 (2019).
4. Jiang J, Zhou H, Jiang Q, Sun L, Deng P. Novel transforming growth factor-beta receptor 1 antagonists through a pharmacophore-based virtual screening approach. *Molecules* **23**, 2824 (2018).
5. Yan G, *et al.* Pharmacophore-based virtual screening, molecular docking, molecular dynamics simulation, and biological evaluation for the discovery of novel BRD 4 inhibitors. *Chemical Biology & Drug Design* **91**, 478-490 (2018).

3. There is some attempt to demonstrate correlations between 2D vs 3D Euclidean distance pharmacophore in the supplementary information. While the authors show that there is a strong pair-wise correlation, a pharmacophore hypothesis usually contains >3 pharmacophore points and here the comparisons from 2D to 3D mapping were pair-wise making it sort of trivial. I would suggest comparing the correlation of all the pharmacophore points together to demonstrate that the original pharmacophores were maintained.

Authors' response:

We agree with your comments. Following your suggestions, we have conducted new experiments, which are shown in the section titled 'Analysis of the correlation and the differences between the shortest-path distance and the Euclidean distance' in Supplementary Information. **Figure R3** illustrates the distribution of the mean absolute differences between the shortest-path distances and Euclidean distances within each pharmacophore hypothesis. All 3D pharmacophores and their corresponding molecules were obtained from ePharmaLib¹, filtering out molecules too large and pharmacophores that did not match the types used in our original experiment. In total, around 50,000 distance pairs and 4,861

pharmacophore hypotheses were analysed. The average absolute error between the shortest-path distance and Euclidean distance is 0.628 Å, which is acceptable relative to the pharmacophore radius of 1.5 Å (the length of a typical C-C bond)². **Figure R4** illustrates the matching degree between the shortest-path distances and Euclidean distances within each pharmacophore hypothesis. The matching degree is calculated as the percentage of distance pairs whose differences are less than 1.5Å. The results show that most of the matching degrees are above 0.9, despite the fact that the level of inconsistency increases when the number of pharmacophore points increases.

Figure R3 (Figure S2). Mean absolute differences between Euclidean distances and shortest-path distances. All 3D Euclidean distance-based pharmacophores and corresponding molecules are from ePharmaLib¹. In total, approximately 50,000 distance pairs and 4,861 pharmacophore hypotheses were used.

Figure R4 (Figure S3). Distribution of the matching degrees between the shortest-path distances and Euclidean distances within each pharmacophore hypothesis. Bars of the same colour indicate a specific number of pharmacophore elements within a hypothesis. The matching degree is calculated as the percentage of distance pairs whose difference is less than 1.5Å (the length of a typical C-C bond). All 3D Euclidean distance-based pharmacophores and corresponding molecules from ePharmaLib. In total, around 50,000 distance pairs and 4,861 pharmacophore hypotheses were used.

Reference

1. Moubrock AF, *et al.* ePharmaLib: A Versatile Library of e-Pharmacophores to Address Small-Molecule (Poly-) Pharmacology. *Journal of Chemical Information and Modeling* **61**, 3659-3666 (2021).
2. Kohlbacher SM, Langer T, Seidel T. QPHAR: quantitative pharmacophore activity relationship: method and validation. *Journal of cheminformatics* **13**, 1-14 (2021).

4. Although it may not be necessary, it could further strengthen the impact of the paper if the authors could test 1-2 predicted compounds for experimental target binding from figure 3b.

Authors' response:

Thank you for this suggestion. We agree that it would be ideal to experimentally validate our predictions. However, most computational methods resort to existing data or docking simulations for validation¹⁻⁶. In our paper, we have shown

that PGMG can generate drug-like molecules with strong docking scores similar to experimentally validated molecules. In the exemplary cases, we have also demonstrated the application of PGMG in different drug design scenarios with different types of activity data. These results suggest that PGMG provides a promising drug design method. During the revision stage, we were trying to consult with a company for molecule synthesis, but there were logistic challenges because of the COVID-19 restrictions in China. Therefore, we decided to validate our predictions in future work. On the other hand, we developed an easy-to-use web server of PGMG that allows users to generate molecules for any given pharmacophore hypotheses. It is available at <https://www.csuligroup.com/PGMG>. We will update our experimental validation results on the web page in the future as well.

Reference

1. Méndez-Lucio O, Baillif B, Clevert D-A, Rouquié D, Wichard J. De novo generation of hit-like molecules from gene expression signatures using artificial intelligence. *Nature communications* **11**, 1-10 (2020).
2. Gebauer NW, Gastegger M, Hessmann SS, Müller K-R, Schütt KT. Inverse design of 3d molecular structures with conditional generative neural networks. *Nature communications* **13**, 1-11 (2022).
3. Mahmood O, Mansimov E, Bonneau R, Cho K. Masked graph modeling for molecule generation. *Nature communications* **12**, 1-12 (2021).
4. Wang M, *et al.* RELATION: a deep generative model for structure-based de novo drug design. *Journal of Medicinal Chemistry* **65**, 9478-9492 (2022).
5. Peng X, Luo S, Guan J, Xie Q, Peng J, Ma J. Pocket2mol: Efficient molecular sampling based on 3d protein pockets. In: *International Conference on Machine Learning*. PMLR (2022).
6. Uludoğan G, Ozkirimli E, Ulgen KO, Karalı N, Özgür A. Exploiting pretrained biochemical language models for targeted drug design. *Bioinformatics* **38**, ii155-ii161 (2022).

6. The authors performed "in-silico binding" of predicted compounds using AutoDock Vina. However, it is unclear from the supplementary information if the results in figure 3b were based on rigid or flexible docking. If it is the former, it is best to specify given that by default the program performs flexible docking.

Authors' response:

We have added the details in the 'Methods' section, as follows:

'We use the AutoDock Vina to perform semiflexible docking with default parameters, where the flexibility of ligands is considered to dock into a rigid receptor. The central coordinates of the box are calculated as the average coordinates of each heavy atom in the ligand. The size of the box is determined by the size of the ligand in the PDB complex'.

7. Another major limitation of predicted molecules from the current generative models is synthetic feasibility, particularly of those SMILES-based generative models. In addition to the several metrics used to assess the performance in the evaluation, I would also recommend evaluating the synthetic accessibility scores of the predicted compounds. See, for example, <https://jcheminf.biomedcentral.com/articles/10.1186/1758-2946-1-8>

Authors' response:

Thank you for this suggestion. We have added a new experiment (**Table R1, as shown in response #1**) to compare the SA of the molecules generated using different methods. A detailed description of the experiment can be found in the 'Results' section of the revised manuscript. As shown in Table R1, compared with other methods of generating active molecules, PGMG obtained superior results in terms of SA.

The synthetic accessibility is denoted as SA in our original manuscript ('Results' section Figure 4b), and we have expanded the evaluation of SA from four targets (including 4,000 molecules) to 15 targets (including 15,000 molecules). As can be seen in **Figure R5**, most of the generated molecules have SA values less than 4, indicating that the molecules generated by PGMG have the potential for simpler synthesis.

Figure R5 (Figure 4b) Distributions of the ADMET properties predicted using ADMETlab 2.0¹ of the top 1,000 molecules generated by PGMG over 15 targets.

Reference

1. Xiong G, *et al.* ADMETlab 2.0: an integrated online platform for accurate and comprehensive predictions of ADMET properties. *Nucleic Acids Research* **49**, W5-W14 (2021).

8. The authors make some claims that the program can predict novel compounds without/or with limited structural activity data. I think this is an advantage of the ligand-based drug design approach in general but not specific to the PGMG program. While it is impressive that the PGMG program can generate potentially bioactive compounds satisfying any pharmacophore constraints, noted that not all pharmacophore points are essential for binding and will still require extensive QSAR study to validate a pharmacophore hypothesis.

Authors' response:

We agree that ligand-based drug design in general requires limited data on the receptor's structure. However, other deep learning-based generative methods generally require (1) a number of molecules with known activity to fine-tune the parameters for a specific target and/or (2) the bioactive conformation of a ligand. These data are not necessary for PGMG, hence giving it a wider application. In addition, compared with traditional methods, PGMG is capable of rapidly generating a large number of molecules with novel structures that maintain the pharmacophore model.

We also agree that not all pharmacophore points are essential and that an extensive QSAR study will produce a better pharmacophore hypothesis. PGMG is designed to generate drug-like molecules matching a given pharmacophore hypothesis, no matter how it is constructed. One of the strengths of PGMG is that users can construct pharmacophore hypotheses using expert knowledge, thus having more flexibility in the output of PGMG. If extensive QSAR studies are performed to build the hypotheses, the quality of the generated molecules may be further improved. We have added the discussion about this in the 'Discussion' section.

Reviewer #2:

PGMG: A Pharmacophore-Guided Deep Learning Approach for Bioactive Molecular Generation (Zhu et al.)

The authors propose a method to condition SMILES generation with pharmacophores. They follow traditional VAE methods to embed SMILES in a latent space using Transformer encoders/decoders, and pre-process pharmacophores as fully connected graphs between the spatial features into a conditioning vector that will be concatenated to the SMILES embedding for encoding or latent space vector for decoding. They show that their method reaches satisfactory validity, uniqueness and novelty of generated molecules compared to other popular SMILES generative models, while reaching high matching to conditioning pharmacophore. They further demonstrate that generated molecules using structure-based pharmacophore have on average better binding affinity as estimated by AutoDock Vina, while being in acceptable ranges of predicted ADMET properties. These putative high affinity molecules tend to adopt similar binding modes to active molecules or find new interactions. They also show examples of ligand-based pharmacophore matching for terbinafine. Finally, they use the model to generate compounds for an EGFR modified pharmacophore to increase logP of the initial active Lavendustin A, and show that they retrieve molecules similar to other actives that were not included in training.

Overall this is interesting work that, due to the focus of using pharmacophores for design, is also largely novel (though one key comparison to a related method is missing which is close, see below).

Authors' response:

We thank the reviewer for the positive comments. We provide the point-by-point responses below.

Major items:

- This work does not compare nor cite the RELATION work that also used pharmacophores as input for the model: <https://pubs.acs.org/doi/full/10.1021/acs.jmedchem.2c00732>. Can a common benchmark be used to establish relative performance of methods?

Authors' response:

Following your suggestion, we have added a comparison of PGMG with RELATION¹ and two other methods (Pocket2Mol² and Seq2Seq³) aimed at generating bioactive molecules. To the best of our knowledge, there is no commonly used benchmark for evaluating active molecule generation methods. In the comparison experiments, we used these methods to generate 10,000 molecules for AKT1 and CDK2, evaluating the performance using 1) ratio of available molecules, 2) average synthetic accessibility score (SA), 3) average docking score, 4) average alignment score between generated molecules and pharmacophore hypotheses (pharmacophore score) and 5) computational time. The results are shown in Table R1.

Table R1. The experimental results of PGMG and other methods for generating bioactive molecules.

Target	Method	Validity ↑	Uniqueness ↑	Novelty ↑	Ratio of available molecules↑	Mean SA↓	Docking Score↓	Pharmacophore Score↑	Time↓
CDK2	PGMG	0.981	0.949	0.995	92.6%	2.66	-8.68	0.782	18s
	RELATION _{phar} *	0.361	1	1	36.1%	2.86	-8.32	0.742	5s
	RELATION _{phar-} BO _{dock} *	0.622	0.992	0.942	58.1%	2.87	-8.30	0.742	~60h
	Pocket2Mol	1	0.248	0.998	24.8%	4.28	-8.38	-	1.5h
	Seq2Seq	0.953	0.796	0.999	75.8%	2.87	-8.69	-	97s
	PGMG	0.996	0.848	0.993	83.9%	2.45	-10.84	0.762	15s
	RELATION _{phar} *	0.332	1	1	33.2%	3.11	-9.85	0.790	5s

AKTI	RELATION _{phar} - BO _{dock} *	0.541	0.94	0.951	48.4%	3.10	-9.83	0.813	~60h
	Pocket2Mol	1	0.313	0.997	31.2%	4.25	-10.40	-	1.9h
	Seq2Seq	0.945	0.67	1	63.3%	3.20	-10.43	-	103s

*The validity, uniqueness and novelty of two RELATION methods are taken from the original paper. Because different methods have different numbers of available molecules, the synthetic accessibility scores (SA), docking score and pharmacophore score are calculated as the average of the top 2,000 molecules sorted based on their docking scores.

RELATION¹ is a 3D-based generative model that utilises the protein-ligand complex structure and pharmacophore hypotheses of a given target to design novel active-like molecules. A docking-based Bayesian sampling was applied to improve its performance (RELATION_{phar}-BO_{docking}). Compared with PGMG, the pharmacophore hypotheses are fixed in the training stage, which requires further fine-tuning for a given target.

Pocket2Mol² is a graph-based, E(3)-equivariant generative network that can efficiently generate drug-like molecules conditioned on the pocket of a target.

Seq2Seq³ exploits a pretrained biochemical language model with two-stage fine-tuning to generate active-like molecules using the target protein sequence as the input.

Table R1 shows that PGMG achieved the best ratio of available molecules and a top docking score. We also compare the average alignment score between the given 3D pharmacophore hypotheses and generated molecules as the pharmacophore score. This is calculated as the ratio of the overlapping volume between the given pharmacophore elements and reference to the volume of reference pharmacophore elements. We also found that PGMG had a pharmacophore score similar to RELATION, suggesting a correlation between the shortest-path-based pharmacophore and Euclidean distance-based pharmacophore. In terms of the ratio of available molecules, SA score and docking score, PGMG shows substantial improvements compared with RELATION.

Reference

1. Wang M, *et al.* RELATION: a deep generative model for structure-based de novo drug design. *Journal of Medicinal Chemistry* **65**, 9478-9492 (2022).
2. Peng X, Luo S, Guan J, Xie Q, Peng J, Ma J. Pocket2mol: Efficient molecular sampling based on 3d protein pockets. In: *International Conference on Machine Learning*. PMLR (2022).
3. Uludođan G, Ozkirimli E, Ulgen KO, Karalı N, Özgür A. Exploiting pretrained biochemical language models for targeted drug design. *Bioinformatics* **38**, ii155-ii161 (2022).

- As is often the case in the area there is no experimental validation of the generated molecules – this would greatly increase the practical relevance of the work (I leave it up to the editor to judge whether purely computational approaches and validation are of sufficient interest to the journal, this is a frequent difference between computer science and life science thinking)

Authors' response:

Thank you for this suggestion. Just as the reviewer mentioned, the previously published computational methods often have no experimental validation¹⁻⁶. In our study, we have shown that PGMG can generate drug-like molecules with strong docking scores similar to experimentally validated molecules. In the exemplary cases, we have also demonstrated the application of PGMG in different drug design scenarios with different types of activity data. These results suggest that PGMG can be a promising drug design method. During the revision stage, we were trying to consult a company for molecule synthesis, but there were logistic challenges because of the COVID-19 restrictions in China. Therefore, we decided to validate our predictions in future work. That being said, we developed an easy-to-use web server of PGMG that allows users to generate molecules for any given pharmacophore hypotheses. It is available at <https://www.csuligroup.com/PGMG>. We will update our experimental validation results on the web page in the future as well.

Reference

1. Méndez-Lucio O, Baillif B, Clevert D-A, Rouquié D, Wichard J. De novo generation of hit-like molecules from gene expression signatures using artificial intelligence. *Nature communications* **11**, 1-10 (2020).
2. Gebauer NW, Gastegger M, Hessmann SS, Müller K-R, Schütt KT. Inverse design of 3d molecular structures with conditional generative neural networks. *Nature communications* **13**, 1-11 (2022).
3. Mahmood O, Mansimov E, Bonneau R, Cho K. Masked graph modeling for molecule generation. *Nature communications* **12**, 1-12 (2021).
4. Wang M, *et al.* RELATION: a deep generative model for structure-based de novo drug design. *Journal of Medicinal Chemistry* **65**, 9478-9492 (2022).
5. Peng X, Luo S, Guan J, Xie Q, Peng J, Ma J. Pocket2mol: Efficient molecular sampling based on 3d protein pockets. In: *International Conference on Machine Learning*. PMLR (2022).
6. Uludoğan G, Ozkirimli E, Ulgen KO, Karalı N, Özgür A. Exploiting pretrained biochemical language models for targeted drug design. *Bioinformatics* **38**, ii155-ii161 (2022).

- This is also reflected in the wording – the authors need to make clear that properties are predicted, interactions putative (if based on docking), etc. E.g. "To further assess the pharmacokinetics and toxicity of the generated molecules, we calculate the TSPA, 280SA, and hERG of the generated molecules." (line 280) – PK and toxicity get assessed by in vivo experiments, not by 'calculating TSPA (TPSA?), SA and hERG' activity (etc, throughout text)

Authors' response:

Thank you for pointing this out. We have carefully examined the entire text and modified these descriptions in the revised manuscript. Furthermore, we have proofread the manuscript, and corrected grammatical errors using a professional language editing service.

- The shortest-path proxy sounds great, but is there any cases where it fails? For example, in the scatterplot Figure S1, we can see points with low Euclidean distance (< 4) and high shortest-path distance (>8), likely to correspond to "folded" conformations. I would expect the model to "succeed" in having this shortest-path matching, but failing at the 3D Euclidean distance matching. Also, 1000 molecules for Figure S1 seems a bit low to evaluate the rate of the distance mismatches.

Authors' response:

When a molecule is considered in a folded conformation, as demonstrated in **Figure R6**, the shortest-path distances and Euclidean distances between pharmacophore elements cannot match. When the generated molecules have low flexibility, such as **Figure R2 (n)**, the molecules that match shortest-path-based constraints may not align well with 3D pharmacophore hypotheses.

Figure R6 (Figure S4) Alignment conformations of reference molecules (white) and generated molecules (green) with the pharmacophore hypotheses. (a), (b) and (c) represent different pharmacophore hypotheses, where the conformations of the reference molecules are obtained from the PDB complexes (a) 2IRZ, (b) 2Y1W and (c) 3JZK. The coloured spheres represent different pharmacophore elements: aromatic ring (red), hydrophobic group (green), hydrogen bond donor (purple) and hydrogen bond acceptor (blue).

Figure R2 (Figure 5) Display of the binding sites of PGMG-generated molecules in structure-based drug design. The binding sites and the pharmacophore hypothesis are highlighted within the four receptors, along with some generated molecules. The different coloured spheres represent different pharmacophore features: aromatic ring (red), hydrophobic group (green), hydrogen bond donor (purple) and hydrogen bond acceptor (blue). The grey molecules represent reference molecules in the crystal structure, and the green molecules represent the molecules generated by PGMG.

However, a molecule that matches shortest-path-distance-based pharmacophore constraint may also successfully bind to targets, even if it is in a folded conformation and not perfectly aligned with 3D pharmacophore. For example, in **Figure R2** (i-k), despite the Euclidean distance between the aromatic ring and lower hydrogen bond acceptor failing to match the shortest-path distance, the generated molecules (**Figure R2** (i-k)) can bind to the target and obtain a good docking score.

As for Figure S1, we agree that 1,000 molecules may not be sufficient. To address your concern, we have updated Figure S1 in the manuscript (see **Figure R7** below) using all the pharmacophores and their corresponding molecules from ePharmaLib¹, filtering out molecules too large and pharmacophores that did not match the type used in our original experiment. In total, around 50,000 distance pairs and 4,861 pharmacophore hypotheses were used. As illustrated in **Figure R7**, the shortest-path distances between pharmacophore features are strongly correlated with the Euclidean distances, here with a Pearson correlation coefficient of 0.926.

Figure R7 (Figure S1) Euclidean and mapping distances between pharmacophore features. In total, around 50,000 distance pairs

and 4,861 pharmacophore hypotheses were used. All 3D Euclidean distance-based pharmacophores and corresponding molecules from ePharmaLib. The color bar represents the probability density calculated using a Gaussian kernel density estimation function.

Reference

1. Moubrock, A.F. et al. ePharmaLib: A Versatile Library of e-Pharmacophores to Address Small-Molecule (Poly-) Pharmacology. *Journal of Chemical Information and Modeling* **61**, 3659-3666 (2021).

- In the ablation study, what exactly is random sampling? Is it uniform sampling or something else?

Authors' response:

The *random sampling* refers to a technique used in the generation process of SMILES. In the generation process, the model will produce a discrete probability score distribution of the next token, given the latent variable z and formerly generated tokens. By default and for stability, the next token chosen by PGMG is the token with the highest score. That being said, the next token can also be chosen by sampling the multinomial distribution defined by the scores, which is random sampling in the ablation study. We have incorporated the description of the random sampling in the revised Supplementary Information ('Ablation study' section).

- The training data uses random pharmacophores extracted from molecules. Does it include active compounds towards the target used in the structure-based pharmacophore? For instance, Figures 4a to c shows very similar scaffolds, is there a way that the model has seen a compound that shows a more or less specific pharmacophore of VEGFR2 before?

Authors' response:

Yes, the training data include the reference ligand and molecules similar to the reference ligand, so there is a possibility that similar pharmacophores have been seen by the model during training. However, considering how large the total number of molecules and how small the probability is, the effect of these molecules on the generation result is negligible.

In our paper, the ChEMBL 24 dataset used to train PGMG contains more than 1.25 million drug-like molecules, and as a result, we find that the ligand in the PDB structure 1YWN (VEGFR2) is indeed in the training set. We further examined the training data and found that only 205 molecules in the dataset had both a Tanimoto similarity and MACCSkeys fingerprints greater than 0.8 compared to the reference ligand in 1YWN. The average similarity between the displayed molecules and the reference molecule was 0.87. These 205 molecules, on average, have 18 pharmacophore elements, and the probability of choosing the same pharmacophore as VEGFR2 to form a training sample that leaks is about $1/C_{18}^6 \approx 5.4 * 10^{-5}$. Considering how large the total number of molecules is and how small the probability is, these molecules have a negligible effect on the generation results of PGMG.

From another perspective, one of the reasons that the generated molecules (**Figure R2 (a-c)**) and reference molecules (**Figure R2 (d)**) share similar scaffolds may be that the pharmacophore hypothesis used to generate these molecules has more elements than other targets (**Figure R2**), putting a stronger restriction on the scaffolds. In the region without the restriction of pharmacophore points, the generated molecules have significant differences compared with the reference molecules.

- In the structure-based pharmacophore generation, the comparison is made between the top-1000 generated molecule and a variable number of known actives for the target. It would maybe be fair to compare with the top-1000 known actives (given that some low-activity compounds can be a result of exploring SAR, and not just optimization towards higher activity).

Authors' response:

Thank you for this suggestion. We have updated the manuscript by comparing the top 1,000 generated molecules with the top 1,000 molecules with known activity. We have expanded the scale of this experiment in the revised manuscript from 4 to 15 targets. The results are given in **Figure R1**, and the details are described below.

We obtained pharmacophore hypotheses with known target structures from the literature and kept those targets with more than 1500 active molecules. For each pharmacophore model, 10,000 molecules were generated by PGMG. AutoDock Vina, with the default parameters, was used to predict the binding affinities of generated molecules. In **Figure R1**, we show the affinity distributions of the top 1,000 molecules generated by PGMG and top 1,000 affinities for the known bioactivity molecules from ChEMBL. The docking scores for the two sets of molecules are similar, suggesting that these PGMG-generated molecules can bind to the 3D structure of the targets.

Figure R1 (Figure 4a) Box plot of docking scores of the top 1,000 molecules generated by PGMG and top 1,000 molecules collected with known bioactivity over 15 targets.

- On the ligand-based pharmacophore for terbinafine, the authors mention that they obtained the active conformation of terbinafine from Drugbank. Is it really the bioactive conformation (from experimental determination), or is it a low-energy conformation? If bioactive, then a comparison could be made with the determined target, if not, then the extracted pharmacophore might not reflect the one of the bound pose.

Authors' response:

The reference conformation in DrugBank was previously a low-energy conformation, but because of the lack of an experimentally determined active conformation of terbinafine, we use a conformation obtained from literature¹ in the revised manuscript ('Demonstration of PGMG's application in Ligand-based drug design'). The new alignment results are shown in **Figure R8**. The conformation of terbinafine was determined by a molecular dynamics simulation with a fully atomic 3D model of squalene epoxidase from *S. cerevisiae*. The paper concludes that, in terbinafine, the tertiary amine, the aromatic ring and the hydrophobic group centred on tert-butyl play a major role in binding to squalene epoxidase, which is also supported by the analysis of human squalene epoxidase (PDB ID 6C6P) and a terbinafine analogue². The above analysis shows that the extracted pharmacophore reflects the binding mode of squalene epoxidase and terbinafine. For ligand-based drug design scenarios, it is often difficult to obtain the active conformation of the reference molecule. One of the advantages of PGMG is that the active conformation is not required for the construction of the pharmacophore, so using the shortest-path pharmacophore restriction generates a molecule that is consistent with the pharmacophore of the reference molecule.

Figure R8 (Figure 6) Alignment of terbinafine (grey) and molecules (green) generated by PGMG. The coloured spheres represent different pharmacophore elements, including aromatic ring (red), cation (yellow) and hydrophobic group (green).

References

1. Nowosielski, M. et al. Detailed mechanism of squalene epoxidase inhibition by terbinafine. *Journal of Chemical Information and Modeling* **51**, 455-462 (2011).
2. Padyana AK, et al. Structure and inhibition mechanism of the catalytic domain of human squalene epoxidase. *Nature Communications* **10**, 97 (2019).

- On the modified pharmacophore of Lavendustin A, the authors only test among the compounds with perfectly matching pharmacophore the LogP and similarity to known actives, which are convincing candidates. Another validation could be docking to known EGFR structure to try and evaluate if the binding mode is similar.

Authors' response:

We have added the binding mode analysis in the revised Supplementary Information ('Analysis of the correlation and the differences between the shortest-path distance and the Euclidean distance'), as follows:

We took the crystal structure (PDB ID: 1M17) of EGFR and used AutoDock Vina to predict the binding pose for Lavendustin A and the generated molecules (**Figure R9**). The coordinates of the highlighted 3D pharmacophore points were determined as the corresponding atom coordinates in the docking conformation of Lavendustin A. The generated molecules have the same aromaticity, hydrophobic region (Leu694, Lys721) and hydrophilic region (Met769) as Lavendustin A. This implies that the generated molecules can have the same binding mode restricted by the given pharmacophore as that of Lavendustin A¹.

Figure R9 (Figure S9) The binding surface of Lavendustin A and the generated molecules in the pocket of EGFR (1M17). (a) Lavendustin A, (b-f) generated molecules by PGMG. The different coloured spheres represent different pharmacophore features: aromatic ring (red), hydrogen bond donor (purple) and hydrophobic group (green). Lys721 (K721), Leu764 (L764), Leu694(L694) and Met769(M769) represent amino acid residues from the crystal structure of EGFR (1M17).

Reference

1. Żolek, T., Trzeciak, A. & Maciejewska, D. Theoretical evaluation of EGFR kinase inhibition and toxicity of di-indol-3-yl disulphides with anti-cancer potency. *Journal of Biomolecular Structure and Dynamics* **40**, 622-634 (2022).

- A missing measure in this work (and lots of other generative modelling research in general) is the generation time. For instance, PGMG is compared to popular SMILES generator, reaching satisfactory metrics. However, if current work generates 1000 molecules per second while the VAE approach generated 10000 molecules, you will obtain a higher number of valid, unique and novel (that the authors refer to as available) molecules in the same amount of time.

- How does the current work compare with SMILES generator conditioned on docking score. I would expect PGMG to be faster on the generation, but I wonder what the difference between the predicted binding affinity range and match score range would be.

Authors' response:

In the revised manuscript, we have compared the generation time, the docking score and other metrics with several other methods (RELATION¹, Pocket2Mol² and Seq2Seq³) that also aim at generating molecules with activity. The results are shown in **Table R1**, here indicating that PGMG has an advantage in terms of generation time. Under the same conditions, PGMG generates 10,000 molecules in 18 s, which is faster than the Seq2Seq (97 s) model, which takes the sequence as input, and is significantly faster than the RELATION_{phar-BO_{dock}*} (60 h) and Pocket2Mol (1.5 h).

The PGMG achieves the best docking score for AKT1 and the second best for CDK2, only 0.01 behind Seq2Seq. We also compare the matching degree between the generated molecules and original 3D pharmacophore hypotheses, here denoted as the pharmacophore score, with RELATION. Surprisingly, compared with RELATION, which directly encodes the 3D pharmacophore hypotheses, PGMG yields a comparable average pharmacophore score. From **Table R1**, it can be seen that the PGMG and RELATION methods both achieve good results in their pharmacophore and docking scores.

Table R1. The experimental results of PGMG and other methods for generating bioactive molecules.

Target	Method	Validity ↑	Uniqueness ↑	Novelty ↑	Ratio of available molecules ↑	Mean SA ↓	Docking Score ↓	Pharmaco phore Score ↑	Time ↓
CDK2	PGMG	0.981	0.949	0.995	92.6%	2.66	-8.68	0.782	18s
	RELATION _{phar} *	0.361	1	1	36.1%	2.86	-8.32	0.742	5s
	RELATION _{phar} - BO _{dock} *	0.622	0.992	0.942	58.1%	2.87	-8.30	0.742	~60h
	Pocket2Mol	1	0.248	0.998	24.8%	4.28	-8.38	-	1.5h
	Seq2Seq	0.953	0.796	0.999	75.8%	2.87	-8.69	-	97s
AKT1	PGMG	0.996	0.848	0.993	83.9%	2.45	-10.84	0.762	15s
	RELATION _{phar} *	0.332	1	1	33.2%	3.11	-9.85	0.790	5s
	RELATION _{phar} - BO _{dock} *	0.541	0.94	0.951	48.4%	3.10	-9.83	0.813	~60h
	Pocket2Mol	1	0.313	0.997	31.2%	4.25	-10.40	-	1.9h
	Seq2Seq	0.945	0.67	1	63.3%	3.20	-10.43	-	103s

*The validity, uniqueness and novelty of two RELATION methods are taken from the original paper. Because different methods have different numbers of available molecules, the synthetic accessibility scores (Mean SA), docking score and pharmacophore score are calculated as the average of the top 2,000 molecules sorted based on their docking scores. ‘↑’ means the higher the value, the better, whereas ‘↓’ means the lower the value, the better.

Reference

1. Wang M, *et al.* RELATION: a deep generative model for structure-based de novo drug design. *Journal of Medicinal Chemistry* **65**, 9478-9492 (2022).
2. Peng X, Luo S, Guan J, Xie Q, Peng J, Ma J. Pocket2mol: Efficient molecular sampling based on 3d protein pockets. In: *International Conference on Machine Learning*. PMLR (2022).
3. Uludoğan G, Ozkirimli E, Ulgen KO, Karalı N, Özgür A. Exploiting pretrained biochemical language models for targeted drug design. *Bioinformatics* **38**, ii155-ii161 (2022).

- As you mention, including exclusion volume in pharmacophore is a relevant perspective. It would be great to see some examples where those could be efficient, as I expect the current approach to maximize the match score but at the cost of not being able to bind in the real pocket *without clashes (cf. remark on docking compounds for EGFR pharmacophore)

Authors' response:

As shown in **Figure R10 (b-c)**, when considering only the alignment with a given pharmacophore hypothesis, without taking into account protein pockets, the conformations of the generated molecules conflicted with the structure of the pocket. As a result, the docking conformations in **Figure R10 (e-f)** do not preserve the original binding mode. This issue can be avoided with the help of the exclusion volume. On the other hand, we have found that small drug-like molecules generated by PGMG often inherently satisfy the exclusion volume constraints (**Figure R9**). Still, including the exclusion volume is expected to increase the quality of the generated molecules, especially for large ones.

Figure R10. Examples of the effect of exclusion volume. (a)(d) The reference ligand (Lavendustin A) and its binding conformation. (b)(c) Conformations of two generated molecules that can match the pharmacophore hypothesis but would clash with the pocket. (e)(f) Docking conformation of the two generated molecules. The different coloured spheres represent different pharmacophore features: aromatic ring (red), hydrogen bond donor (purple) and hydrophobic group (green).

Minor comments:

1.10: bioactive instead of bioactivate

1.120: pharmacophore instead of pharamacophore

1.214: calculates seems optimistic, Autodock vina only predicts/estimates the binding affinity

1.238: could be rephrased: "share interaction with the same amino acid residues"

1.280 and 1.282: TPSA and not TSPA

1.284: Figure 5 should be bold font to match previous formatting

1.362: Would be easier for the reader to directly have the PDB ids in the methods, to avoid going back to the results section

1.452: novel molecules instead of novelty molecules

Detailed comments on the supplementary information:

1.9: Basefeatures.fdef and not Basefeatures.fdef. Might be great to add a direct link as well from the official rdkit github repo to know where to find it

1.16: feature instead of features

1.42: a node set instead of anode set

Authors' response:

We have corrected these grammar errors and typos. Furthermore, we have asked a professional language editing service (Scribendi service) to proofread the whole manuscript.

Reviewer #3:

This paper presents a deep-learning based approach to generate molecules with matched pharmacophore structures. Technically, the method appears to be interesting and has its own value, there are serious concerns about the evaluation of the method. Thus, I'm not fully convinced that the paper is novel and efficient enough to be published in Nature Communications.

Authors' response:

We thank the reviewer for these critical comments. We have added additional comparison methods and conducted more experiments to improve the evaluation of PGMG. To improve the efficiency of using PGMG, we developed an easy-to-use web server of PGMG that allows users to generate molecules for any given pharmacophore hypotheses. It is available at <https://www.csuligroup.com/PGMG>. Please find the point-by-point responses below.

1. The ratios of valid and novel compounds are compared with ORGAN, VAE, SMILES LSTM, and Syntalinker. Most compared methods are published a while ago and many more efficient methods are introduced. The comparison with relatively early models is not fully convincing.

Authors' response:

We have included several recently published methods aimed at generating active-like molecules (RELATION, Pocket2Mol² and Seq2Seq) compared with PGMG in the revised manuscript. RELATION is a 3D-based generative model that utilises the protein-ligand complex structure and pharmacophore hypotheses of a given target to design novel active-like molecules. A docking-based Bayesian sampling was also applied to improve its performance. Compared with PGMG, the pharmacophore hypotheses are fixed in the training stage and require further fine-tuning for a given target. Pocket2Mol is a graph-based E(3)-equivariant generative network that can efficiently generate drug-like molecules conditioned on the pocket of a target. Seq2Seq refers to the method developed by Uludoğan et al., which exploits a pretrained biochemical language model with a two-stage fine-tuning process to generate active-like molecules using the target protein sequence as the input.

Because generating novel molecules without other restrictions is relatively easy and a simple rule-based programme could achieve this goal, we pay more attention to evaluating these metrics in generating active-like molecules. Generating molecules with activity is much more difficult and more of a real-world problem⁴. In **Table R1**, we find that PGMG performs the best overall, with a significantly improved ratio of available molecules and almost the best docking score. Also, PGMG has an advantage in terms of generation time. For example, in the generation of bioactive molecules towards CDK2, PGMG generates 10,000 molecules in 18 seconds, which is faster than the Seq2Seq model (97 s), and significantly faster than the RELATION_{phar}-BO_{dock} (~60 h) and Pocket2Mol (1.5 h), only a few seconds behind RELATION_{phar}. Besides, in terms of synthetic accessibility score, PGMG shows substantial improvements over other methods.

Table R1. The experimental results of PGMG and other methods for generating bioactive molecules.

Target	Method	Validity ↑	Uniqueness ↑	Novelty ↑	Ratio of available molecules ↑	Mean SA ↓	Docking Score ↓	Pharmaco phore Score ↑	Time ↓
CDK2	PGMG	0.981	0.949	0.995	92.6%	2.66	-8.68	0.782	18s
	RELATION _{phar} *	0.361	1	1	36.1%	2.86	-8.32	0.742	5s
	RELATION _{phar} - BO _{dock} *	0.622	0.992	0.942	58.1%	2.87	-8.30	0.742	~60h
	Pocket2Mol	1	0.248	0.998	24.8%	4.28	-8.38	-	1.5h
	Seq2Seq	0.953	0.796	0.999	75.8%	2.87	-8.69	-	97s
AKT1	PGMG	0.996	0.848	0.993	83.9%	2.45	-10.84	0.762	15s

RELATION _{phar} *	0.332	1	1	33.2%	3.11	-9.85	0.790	5s
RELATION _{phar-} BO _{dock} *	0.541	0.94	0.951	48.4%	3.10	-9.83	0.813	~60h
Pocket2Mol	1	0.313	0.997	31.2%	4.25	-10.40	-	1.9h
Seq2Seq	0.945	0.67	1	63.3%	3.20	-10.43	-	103s

*The validity, uniqueness and novelty of two RELATION methods are taken from the original paper. Because different methods have different numbers of available molecules, the synthetic accessibility scores (Mean SA), docking score and pharmacophore score are calculated as the average of the top 2,000 molecules sorted based on their docking scores. ‘↑’ means the higher the value, the better, whereas ‘↓’ means the lower the value, the better.

Reference

1. Wang, M. et al. RELATION: A deep generative model for structure-based de novo drug design. *Journal of Medicinal Chemistry* **65**, 9478-9492 (2022).
2. Peng X, Luo S, Guan J, Xie Q, Peng J, Ma J. Pocket2mol: Efficient molecular sampling based on 3d protein pockets. In: *International Conference on Machine Learning*. PMLR (2022).
3. Uludoğan, G., Ozkirimli, E., Ulgen, K.O., Karalı, N. & Özgür, A. Exploiting pretrained biochemical language models for targeted drug design. *Bioinformatics* **38**, ii155-ii161 (2022).
4. Tripp, A., Chen, W. & Hernández-Lobato, J.M. An evaluation framework for the objective functions of de novo drug design benchmarks. In: *ICLR2022 Machine Learning for Drug Discovery* (2022).

2. The improvements in binding affinities, based on the Autodock-vina score, are tested with only four targets. It is not clear whether this improvement is general and transferrable to other targets. More comprehensive tests are necessary.

Authors' response:

We completely agree with your comments. Following your suggestions, we conducted more experiments by increasing the number of targets from 4 to 15.

We obtained pharmacophore models from the literature with known target structures, keeping those targets with more than 1,500 known active molecules. For each pharmacophore model, 10,000 molecules were generated by PGMG and AutoDock Vina was used to predict the binding affinities of generated molecules. We compared the docking score between the top 1,000 generated molecules and top 1,000 known active molecules. Next, the top 1,000 molecules with the strongest docking score were compared with the top 1,000 molecules with the strongest binding affinity. In **Figure R1**, we show the docking score distributions of the generated molecules and known bioactive molecules collected from ChEMBL. The docking scores of the top 1,000 molecules generated by PGMG are generally comparable with the top 1,000 active molecules in the ChEMBL database, demonstrating the ability of PGMG to generate active-like molecules.

Figure R1 (Figure 4a). Box plot of docking scores of the top 1,000 molecules generated by PGMG and top 1,000 molecules collected with known bioactivity over 15 targets.

3. The result shown in Figure 3a is the only result that shows how much generated molecules are consistent with a given pharmacophore restraint. A comparison with random molecules appears to be obvious. More comparisons with existing

molecular generation methods that use pharmacophore information are needed.

Authors' response:

Thank you for this suggestion. Since no other generative methods use the same shortest-path-distance-based constraint with PGMG, we can only show the distribution of matching score of PGMG and random molecules. In our experiment, as shown in Figure 3 (**Figure R11**), most of the generated molecules (83.6%) have matching scores greater than 0.8, in which 78.6% have a matching score of 1.0. In contrast, the matching degrees for the random molecules are centred at 0.466, with only 4.91% having a matching score of 1.0. The high level of the matching score demonstrates PGMG's ability to generate molecules satisfying the given pharmacophore hypotheses.

Figure R11 (Figure 3) The distributions of the match scores of PGMG-generated molecules compared with randomly selected molecules.

Furthermore, we have added the comparison of PGMG and a recently proposed method called RELATION¹ in the revised manuscript. The experiment has been mentioned in the response to your first comment. RELATION is a 3D-based generative methods, which, compared with PGMG, uses 3D pharmacophores and has several other requirements to generate active-like molecules towards a target, including a dataset of active ligands and the protein-ligand complex structure, as well as an additional training stage. To the best of our knowledge, we do not find other methods that use pharmacophore hypotheses as the input. As shown in Table R1, PGMG shows substantial improvements in the ratio of available molecules (from 58.1% to 92.6% in CDK2 and 48.4% to 83.9% in AKT1). Although the constraint used by PGMG is a 2D constraint, we also compared the matching degree between the given 3D pharmacophore hypotheses and generated molecules. The matching degree (pharmacophore score) is calculated as the ratio of the overlapping volume between the given pharmacophore elements and reference to the volume of reference pharmacophore elements. Surprisingly, PGMG has pharmacophore scores similar to those of RELATION. Additionally, compared to RELATION, PGMG has better synthetic accessibility scores and docking scores. These results demonstrated the advanced performance of PGMG.

Reference

1. Wang, M. et al. RELATION: A deep generative model for structure-based de novo drug design. *Journal of Medicinal Chemistry* **65**, 9478-9492 (2022).
4. The lead optimization results shown in Figure 6 do not fully support whether this method is efficient for lead optimization. The generated molecules show large variations from the seed molecule, which appears to be too large for lead optimization. Also, the argument, "Based on the assumption that structurally similar molecules have similar properties, the similarity result demonstrates that molecules generated by PGMG have a probability of inhibiting EGFR." Appears to be weak.

4 (a) The lead optimization results shown in Figure 6 do not fully support whether this method is efficient for lead optimization. The generated molecules show large variations from the seed molecule, which appears to be too large for lead optimization.

Authors' response:

We agree that this case differs from the lead optimisation task. The objective here is to design molecules that maintain the activity of Lavendustin A while improving the lipophilicity. Unlike typical lead optimisation tasks, in contrast to directly maintaining similarity with the seed molecule, the pharmacophoric pattern is to be maintained to generate the desired molecules with diverse scaffolds. Rather than 'lead optimisation', it could better be termed 'scaffold hopping'. This is useful to discover molecules that have different ADMET properties with similar binding affinities or to escape from the chemical space covered by existing patents¹. These molecules with novel scaffolds can be further optimised with fine-grained methods. We have made the following modifications in the revised manuscript and revised Supplementary Information to make the description of this case clearer.

'Scaffold hopping refers to the acquisition of molecules with novel scaffolds by replacing the chemical core structure while maintaining some essential features of the known active compounds. It has been widely applied to generate novel backbones to improve physicochemical and ADMET properties or to arrive at patentable analogues. As pharmacophores define the chemical features that are essential for biological activity, they can be employed to guide scaffold replacements. Here, we show how PGMG can help scaffold hopping using Lavendustin A as a case study. Lavendustin A is an inhibitor of epidermal growth factor receptor (EGFR), but it is difficult to cross the cell membrane because of its poor lipophilicity. It has been shown that improving the lipophilicity of Lavendustin A can lead to nanomolar levels of IC₅₀ inhibition activity at the cellular level. We construct a pharmacophore hypothesis using Pharao, and three pharmacophore features are retained by analysing the binding sites of Lavendustin A in the EGFR protein pocket. Then, we use PGMG to generate molecules for the given pharmacophore hypothesis.' ('A showcase of PGMG application in scaffold hopping' in the revised manuscript')

4 (b) Also, the argument, "Based on the assumption that structurally similar molecules have similar properties, the similarity result demonstrates that molecules generated by PGMG have a probability of inhibiting EGFR." Appears to be weak.

Authors' response:

The description has been modified in the revised manuscript to make a clearer statement. The goal of this case study is to identify novel scaffolds that differ from the reference but have the potential to be active. **Figure R12** shows the generated molecules with the same pharmacophore but different scaffolds and their closest EGFR inhibitors obtained from the ExCAPE database². The high similarities with EGFR inhibitors suggest that this goal is successfully achieved.

'Furthermore, we have added new analysis of the binding regions of these generated molecules. These generated molecules have the same aromaticity, hydrophobic region (Leu694, Lys721) and hydrophilic region (Met769) as Lavendustin A³. This implies that the generated molecules can have the same binding mode restricted by the given pharmacophore as Lavendustin A³. The high similarity to the EGFR bioactive molecules and consistent binding site with Lavendustin A of the generated molecules indicates that PGMG can discover these inhibitors that have novel scaffolds with only the knowledge of Lavendustin A.' ('Analysis of the binding sites of the generated molecules' section in the Supplementary Information).

Figure R12 (Figure 7) The molecule generated by PGMG with known inhibitors in the case of scaffold hopping. Molecules generated by PGMG are shown inside the circle, and their closest active nearest neighbours are shown outside the circle. The colours indicate the pharmacophore features extracted from Lavendustin A: aromatic ring (red), hydrogen bond acceptor (blue) and hydrophobic group (green).

Figure R9 (Figure S9) The binding surface of Lavendustin A and the generated molecules in the pocket of EGFR (1M17). (a) Lavendustin A, (b-f) generated molecules by PGMG. The different coloured spheres represent different pharmacophore features: aromatic ring (red), hydrogen bond donor (purple) and hydrophobic group (green). Lys721 (K721), Leu764 (L764), Leu694(L694) and Met769(M769) represent amino acid residues from the crystal structure of EGFR (1M17).

Reference

1. Horvath, D. Pharmacophore-based virtual screening. *Chemoinformatics and Computational Chemical Biology*, 261-298 (2010).
2. Sun, J. et al. ExCAPE-DB: an integrated large scale dataset facilitating Big Data analysis in chemogenomics. *Journal of Cheminformatics* **9**, 1-9 (2017).
3. Żolek, T., Trzeciak, A. & Maciejewska, D. Theoretical evaluation of EGFR kinase inhibition and toxicity of di-

indol-3-yl disulphides with anti-cancer potency. *Journal of Biomolecular Structure and Dynamics* **40**, 622-634 (2022).

REVIEWER COMMENTS

Reviewer #1 (Remarks to the Author):

The authors did an excellent job addressing most of my questions. Below are additional comments:

1. Regarding my previous comments in point 6, the authors confirmed that the docking mode for autodock vina was flexible. I am concerned that the pharmacophore distance could still change after flexible docking (as the docking algorithm could be rather greedy in searching of the best pose). Based on the comments from reviewer #2, the authors did look at the correlation between shortest path distance vs euclidean distance prior docking. Perhaps it would be good to perform another comparison to the 3D pharmacophore after docking?

2. Could the authors provide more details on the scoring function used for the autodock vina? I believe the hydrophobic term is much more dominated than the other interactions. In the docking validation, a generated molecule that docks as good as a known active could simply mean that the two shared similar core (scaffold), which dominated the hydrophobic term but speak little about the interactions of the proposed pharmacophore and if the pharmacophore interaction is similar to the true active.

3. Related to the above points, perhaps the authors can perform rigid docking or better yet pharmacophore-based docking. I believe programs like MOE, Schrodinger and others offer docking modes based on given pharmacophore hypothesis.

Reviewer #3 (Remarks to the Author):

The authors have address all the concerns. They performed the benchmarking with other up-to-date algorithms in various measures.

Also, they expanded the test systems from 4 to 15 targets to show that the binding affinities of the design molecules are comparable to the known inhibitors.

Thus, I believe that the revised manuscript is ready to be accepted.

Response letter

On behalf of all the contributing authors, we would like to express our sincere appreciation for the reviewers' constructive comments and helpful suggestions regarding our article entitled '*PGMG: A Pharmacophore-Guided Deep Learning Approach for Bioactive Molecule Generation*'. In response to the feedback from the reviewers, we have made substantial revisions to our manuscript and have added extra materials to strengthen our results. All the changes have been highlighted in the revised manuscript. We hope that this response letter resolves the reviewers' concerns.

Reviewer #1:

The authors did an excellent job addressing most of my questions. Below are additional comments:

Authors' response:

We thank the reviewer for the positive comment. Please find the point-by-point responses below.

1. Regarding my previous comments in point 6, the authors confirmed that the docking mode for autodock vita was flexible. I am concerned that the pharmacophore distance could still change after flexible docking (as the docking algorithm could be rather greedy in searching of the best pose). Based on the comments from reviewer #2, the authors did look at the correlation between shortest path distance vs euclidean distance prior docking. Perhaps it would be good to perform another comparison to the 3D pharmacophore after docking?

Authors' response:

Thank you for your valuable feedback. We acknowledge your concerns regarding the potential variation in pharmacophore distances after flexible docking. To address this, we conducted an additional experiment to examine the correlation between the shortest-path distance and the Euclidean distance using the docking conformations retrieved from ePharmaLib database[1]. Furthermore, we also repeat the experiment using the experimental conformations retrieved from PDBbind [2] and the low-energy conformations calculated by RDKit [3]. The analysis results of the correlation between the shortest-path distance and the Euclidean distance on the three datasets are shown in **Figure R1**.

The findings from these comparison results further support the original manuscript's conclusion that there is a robust pairwise correlation between shortest-path distances and Euclidean distances across various sources of molecules and conformations. The updated results are also included in the 'Analysis of the correlation and the differences between the shortest-path distance and the Euclidean distance' section of the Supplementary Information.

Figure R1. The correlation between Euclidean and shortest-path distances on molecular conformations obtained from different sources: docking conformations from ePharmaLib (ePharmaLib); active conformations from PDBbind (PDBbind); low energy conformation generated using RDKit (RDKit). (a) Scatter plot of the pairwise correlation between Euclidean and shortest-path distances. The color bar represents the probability density calculated using a Gaussian kernel density estimation function. (b) Mean absolute differences between Euclidean distances and shortest-path distances within each pharmacophore hypothesis.

The Euclidean and shortest-path distances of each pharmacophore feature pair within a pharmacophore hypothesis are illustrated in **Figure R1 (a)**. The Pearson correlation coefficients between two kinds of different distances under ePharmaLib conformations, PDBbind conformations, and RDKit conformations are 0.924, 0.916, and 0.955, respectively. While the distributions are different, the shortest-path distances and Euclidean distances exhibit a strong pairwise correlation under different sources of molecules and conformations. **Figure R1 (b)** displays the distribution of mean absolute differences between shortest-path distances and Euclidean distances within each pharmacophore hypothesis, using the same data from **Figure R1 (a)**. In various sources of conformations (ePharmaLib, PDBbind, and RDKit), the percentage of measurements demonstrating an absolute error less than 1.5 Å between the shortest-path distance and the Euclidean distance are 79.0%, 83.2%, and 96.4%, respectively.

Below is the detailed description of the experiment setting, which has been included in the ‘Analysis of the correlation and the differences between the shortest-path distance and the Euclidean distance’ section of the Supplementary Information in the revised manuscript:

‘For *ePharmaLib*, we obtained 15,148 pharmacophore hypotheses and their corresponding

molecules and conformations from the ePharmaLib database². The pharmacophores were modeled from protein-ligand complex structure initially extracted from the screening-PDB (sc-PDB)⁴. These complexes were then docked and scored with the Glide program⁵, and the pharmacophoric features were selected according to the predicted binding energy terms. In the experiment, hypotheses with pharmacophoric features that did not match the types supported by RDKit were excluded. The shortest path distance and Euclidean distance were computed for each pharmacophore hypothesis. In total, 188,020 distance pairs and 13,056 pharmacophore hypotheses were utilized.

For *PDBbind*, we retrieved 19,442 molecules and molecular conformations from the PDBbind 2020 database. After filtering out files that failed to be read by RDKit, a total of 19,201 molecular conformations were obtained. We acquired a pharmacophore hypothesis for each molecular conformation by proportionally sampling the frequency and type of pharmacophore features, based on their occurrences in the ePharmaLib database. The shortest path distance and Euclidean distance were computed for each pharmacophore hypothesis. In total, 260,864 distance pairs and 19,201 pharmacophore hypotheses were used.

For *RDKit*, we randomly selected 10,000 molecules from the ChEMBL database⁶ and generated conformations using the ETKDG (Etoile's Triangle-Kekulé Distance Geometry) algorithm⁷. For each conformation, the pharmacophore hypothesis, the shortest path distance and Euclidean distance were acquired in a similar manner as *PDBbind*. Altogether, there were 133,361 distance pairs and 10,000 pharmacophore hypotheses.'

References

1. Moumbock AF, et al. ePharmaLib: A Versatile Library of e-Pharmacophores to Address Small-Molecule (Poly-) Pharmacology. *Journal of Chemical Information and Modeling* **61**, 3659-3666 (2021).
2. Liu H, Su M, Lin H-X, Wang R, Li Y. Public data set of protein–ligand dissociation kinetic constants for quantitative structure–kinetics relationship studies. *ACS omega* **7**, 18985-18996 (2022).
3. Landrum G. RDKit: Open-source cheminformatics.

2. Could the authors provide more details on the scoring function used for the autodock vina? I believe the hydrophobic term is much more dominated than the other interactions. In the docking validation, a generated molecule that docks as good as a known active could simply mean that the two shared similar core (scaffold), which dominated the hydrophobic term but speak little about the interactions of the proposed pharmacophore and if the pharmacophore interaction is similar to the true active.

Authors' response:

AutoDock Vina[1] employs an empirical scoring function inspired by the X-score [2]. The predicted binding affinity is calculated as a weighted sum of five terms: three steric terms, a

hydrophobic term, and a hydrogen bonding term. The default weights are determined based on a refined set of PDBbind database. **Figure R2**, taken from ref [1], demonstrates the relationship between the scores of different scoring combinations and the corresponding distances for a specific pair of atoms. Compared to the hydrophobic term, the scoring function is actually more sensitive to the hydrogen bonding term in a certain distance range. Ideally, the five terms of the scoring function should be appropriately balanced to achieve an optimal performance. Given that AutoDock Vina has been widely used in molecular docking and exhibits excellent performance in terms of docking power and scoring power compared to other academic and commercial software [3-5], the balance is likely to be well addressed. Moreover, when using pharmacophore-based docking method to compare the docking scores between the generated molecules and the active molecules, the results are similar to those obtained using Vina. Details regarding the pharmacophore-based docking experiment can be found in the response to comment 3.

Figure R2. The weighted scoring of steric terms alone, or combined with the hydrophobic or H-bonding interactions, using weights from equation (3) [1].

In the following paragraphs we will provide a further description of the AutoDock Vina scoring function. For a comprehensive description of the AutoDock Vina scoring function, we refer the reviewer to the original paper by Trott and Olson [1].

The interaction between each pair of atoms is determined by five terms, which are subsequently weighted and summed up. The steric interactions are identical for all atom pairs, while hydrophobic interactions occur between hydrophobic atoms and hydrogen bonding is considered where applicable. All the interaction functions are truncated at a cutoff distance of $r = 8 \text{ \AA}$. As given in equation (1), the predicted binding affinity is obtained by accumulating scores calculated on each pair of atoms.

$$E = \sum e_{pair}(d) \quad (1)$$

Here, the surface distance d is calculated using Equation (2), where r represents the

interatomic distance, and R_i and R_j represent the radii of the atoms in the given pair.

$$d = r - R_i - R_j \quad (2)$$

For each pair of atoms, the interaction is calculated as:

$$e_{pair}(d) = -0.0356 * Gauss_1(d) - 0.00516 * Gauss_2(d) + 0.840 * Repulsion(d) - 0.0351 * Hydrophobic(d) - 0.587 * Hbond(d) \quad (3)$$

The first three terms of the scoring function represent the steric terms, which primarily consider the electron cloud overlap and resulting repulsive effects between atoms or substances. The fourth term represents the hydrophobic interaction. When the distance is less than 0.5 Å, the interaction is equal to 1. When the distance exceeds 1.5 Å, the interaction is 0. For distances between 0.5 Å and 1.5 Å, the interaction is linearly interpolated based on the distance. The fifth term corresponds to the hydrogen bonding interaction. When the distance is less than -0.7 Å, the hydrogen bonding interaction is 1. When the distance exceeds 1 Å, the interaction is 0. For distances in between, the hydrogen bonding interaction can be obtained through linear interpolation of the distance.

The specific formulas for each term are listed as follows:

$$Gauss_1 = e^{-(d/0.5\text{\AA})^2} \quad (4)$$

$$Gauss_2 = e^{-((d-3\text{\AA})/2\text{\AA})^2} \quad (5)$$

$$repulsion(d) = \begin{cases} d^2, & \text{if } d < 0 \text{ \AA} \\ 0, & \text{if } d \geq 0 \text{ \AA} \end{cases} \quad (6)$$

$$Hydrophobic(d) = \begin{cases} 1, & \text{if } d < 0.5 \text{ \AA} \\ 1.5 - d, & \text{if } 1.5 \text{ \AA} > d > 0.5 \text{ \AA} \\ 0, & \text{if } d \geq 1.5 \text{ \AA} \end{cases} \quad (7)$$

$$Hbond(d) = \begin{cases} 1, & \text{if } d < -0.7 \text{ \AA} \\ \frac{d}{0.7}, & \text{if } 0 \text{ \AA} > d > -0.7 \text{ \AA} \\ 0, & \text{if } d \geq 0 \text{ \AA} \end{cases} \quad (8)$$

References

1. Trott O, Olson AJ. AutoDock Vina: improving the speed and accuracy of docking with a new scoring function, efficient optimization, and multithreading. *Journal of computational chemistry* **31**, 455-461 (2010).
2. Wang R, Lai L, Wang S. Further development and validation of empirical scoring functions for structure-based binding affinity prediction. *Journal of computer-aided molecular design*

- 16, 11-26 (2002).
- Wang Z, *et al.* Comprehensive evaluation of ten docking programs on a diverse set of protein–ligand complexes: the prediction accuracy of sampling power and scoring power. *Physical Chemistry Chemical Physics* **18**, 12964-12975 (2016).
 - Gaillard T. Evaluation of AutoDock and AutoDock Vina on the CASF-2013 benchmark. *Journal of chemical information and modeling* **58**, 1697-1706 (2018).
 - Su M, *et al.* Comparative assessment of scoring functions: the CASF-2016 update. *Journal of chemical information and modeling* **59**, 895-913 (2018).
3. Related to the above points, perhaps the authors can perform rigid docking or better yet pharmacophore-based docking. I believe programs like MOE, Schrodinger and others offer docking modes based on given pharmacophore hypothesis.

Authors' response:

As suggested by the reviewer, we employed Molecular Operating Environment (MOE) [1] for conducting pharmacophore-guided docking experiments. Specifically, we performed these experiments on the generated molecules with a match score greater than 0.85. We determined the top 1000 molecules based on their docking scores. As a comparison, we collected active molecules (pChEMBL > 4) from the ChEMBL database and performed pharmacophore-guided docking experiments (**Figure R4**). In addition, the docking results of known active molecules without pharmacophore guidance were shown as a reference. The docking results demonstrated that the generated molecules by PGMG exhibited overall comparable performance. This indicates that a considerable portion of the generated molecules were able to maintain the pharmacophore interaction and possess good affinity. The experimental results have been included in the 'Pharmacophore-Guided Docking' section of the Supplementary Information in the revised manuscript

Figure R4. Box plots of the docking scores for 15 targets. PGMG (p) represents pharmacophore-guided docking scores of the top 1,000 molecules generated by PGMG. Know Bioactive Molecules (p) represents pharmacophore-guided docking scores of the top 1,000 active molecules (pChEMBL > 4). Known Bioactive Molecules represent the docking results of all active molecules without pharmacophore guidance.

We used the general mode of MOE to perform pharmacophore-guided docking. Molecules

that do not match the given 3D pharmacophore hypothesis or collides with the pocket are filtered out. For active molecules without pharmacophore-guidance, the “Triangle Matcher” method is used for Placement, while other settings remain the same as for the pharmacophore-guided docking. Below is the detailed description of the experiment setting:

1. Protein preparation: The protein preprocessing was facilitated through the application of the QuickPrep in MOE. Water molecules were removed prior to docking for all targets except for BRD4.
2. Pharmacophore preparation: The radius of the pharmacophore was designated at 1.5 Å.
3. Ligand preparation: The ‘wash’ module was utilized to process the molecules and to facilitate the generation of 3D conformations.
4. Docking parameter settings: The binding site of the ligand in the protein-ligand complex was used as the known sites for docking. The placement was executed via the ‘pharmacophore’ method, utilizing the London dG scoring function and resulting in the generation of 30 conformations per search. During the docking process, the GBVI/WSA dG scoring function was employed, and five top poses to be retained.

Reference

1. Corbeil CR, Williams CI, Labute P. Variability in docking success rates due to dataset preparation. *Journal of computer-aided molecular design* **26**, 775-786 (2012).

Reviewer #3:

The authors have address all the concerns. They performed the benchmarking with other up-to-date algorithms in various measures. Also, they expanded the test systems from 4 to 15 targets to show that the binding affinities of the design molecules are comparable to the known inhibitors. Thus, I believe that the revised manuscript is ready to be accepted.

Authors’ response:

Thank you for your positive comments.

REVIEWERS' COMMENTS

Reviewer #1 (Remarks to the Author):

The authors have answered all of my concerns and I have no further comments at this point.